# miR-181a initiates and perpetuates oncogenic transformation through the regulation of innate immune signaling

Matthew Knarr[1,2,3], Rita A. Avelar[2,3,4], Sreeja C. Sekhar[2,3,4], Lily J. Kwiatkowski[1], Michele L. Dziubinski[2,3,4], Jessica McAnulty [2,3,4], Stephanie Skala[3], Stefanie Avril[1,5], Ronny Drapkin [6] & Analisa DiFeo [2,3,4✉]

Genomic instability (GI) predisposes cells to malignant transformation, however the molecular mechanisms that allow for the propagation of cells with a high degree of genomic instability remain unclear. Here we report that miR-181a is able to transform fallopian tube secretory epithelial cells through the inhibition of RB1 and stimulator-of-interferon-genes (STING) to propagate cells with a high degree of GI. MiR-181a targeting of RB1 leads to profound nuclear defects and GI generating aberrant cytoplasmic DNA, however simultaneous miR-181a mediated inhibition of STING allows cells to bypass interferon mediated cell death. We also found that high miR-181a is associated with decreased IFNγ response and lymphocyte infiltration in patient tumors. DNA oncoviruses are the only known inhibitors of STING that allow for cellular transformation, thus, our findings are the first to identify a miRNA that can downregulate STING expression to suppress activation of intrinsic interferon signaling. This study introduces miR-181a as a putative biomarker and identifies the miR-181a-STING axis as a promising target for therapeutic exploitation.

---

[1] Case Comprehensive Cancer Center, Case Western Reserve University, Cleveland, OH 44106, USA. [2] Department of Obstetrics & Gynecology, The University of Michigan, Ann Arbor, MI 48109, USA. [3] Department of Pathology, The University of Michigan, Ann Arbor, MI 48109, USA. [4] The Rogel Cancer Center, The University of Michigan, Ann Arbor, MI 48109, USA. [5] Department of Pathology, Case Western Reserve University, Cleveland, OH 44106, USA. [6] Penn Ovarian Cancer Research Center, Department of Obstetrics and Gynecology, Perelman School of Medicine, University of Pennsylvania, Pennsylvania, PA, USA. ✉email: adifeo@med.umich.edu

High grade serous ovarian cancer (HGSOC) continues to be the most lethal subtype of ovarian cancer[1]. This high mortality is the consequence of two primary barriers: first, there are currently no reliable methods of early detection, therefore the majority of HGSOC patients present with advanced disease. Second, the majority of HGSOC patients develop resistance to platinum-based chemotherapy and there is a lack of targeted treatment alternatives. Together, this leads to a high probability of recurrence and a correspondingly poor prognosis. The primary challenge in overcoming these barriers is that the early development of HGSOC is still poorly understood. As such there is a critical need to identify the key drivers that promote HGSOC tumor initiation at the earliest stages of malignant transformation.

Initially, the site of origin for HGSOC was thought to be derived from transformative stimuli acting on the ovarian surface epithelium or cortical inclusion cysts[2,3]. However, a convincing precursor lesion could not be found on the ovaries. In the early 2000s, putative precursor lesions termed serous tubal intraepithelial carcinoma (STIC) were detected in the surface epithelia of fallopian tubes prophylactically removed from patients harboring high-risk BRCA1/2 mutations[4–10]. Subsequently, an alternative hypothesis proposed that the majority of HGSOCs originate in the fallopian tube fimbria via oncogenic transformation of fallopian tube secretory epithelial cells (FTSECs). Mounting evidence has since accumulated to support this hypothesis including epidemiological studies, genetically engineered mouse models, as well as phylogenetic analysis of HGSOC and its precursor lesions.

The carcinogenesis of HGSOC begins with "p53 signatures," small regions of histologically normal appearing FTSECs that stain strongly for nuclear p53 due to p53 mutation (which is almost universal in HGSOCs)[9,11–13]. P53 signatures undergo subsequent oncogenic alterations to be transformed into STIC which displays all the characteristics of HGSOC in-situ. Current understanding of HGSOC natural history is primarily limited to the p53 signature=>STIC=>HGSOC model with minimal information about what mechanisms facilitate the transition between each stage. The most recent studies of HGSOC tumor evolution confirm that p53 mutation is the earliest detectable mutation in the HGSOC transformation process[9,11–13]. However, after p53 mutation the mechanisms that drive p53 signature=>STIC=>HGSOC transition are still unclear. STIC shows similar levels of genomic instability to HGSOC indicating that the majority of transforming events occur prior to the STIC phase[14]. Transition time alone from STIC to HGSOC is estimated to be at least 7 years, thus there is considerable opportunity to develop effective early detection methods by elucidating the mechanisms of initial HGSOC transformation[13].

In this study, we identify miR-181a as a potent driver of oncogenic transformation in FTSECs. We have previously shown that miR-181a drives metastasis as well as recurrence in advanced stage HGSOC, and shown that it correlates with poor survival outcomes in patients[15,16]. Furthermore, through a pan-cancer analysis of over 10,000 primary tumors representing 38 different cancer types we found that amplification of miR-181a correlated with poor survival (Supplementary Fig. 1A). Given its potent effects on late-stage disease progression, we sought to investigate whether miR-181a could function as an oncomiR in early HGSOC development. We found that miR-181a overexpression alone is sufficient to promote oncogenic transformation and form tumors in vivo. In addition, miR-181a promotes several phenotypes that drive GI. We identified that miR-181a's tumor forming ability was mediated through the cooperative inhibition of the classic tumor suppressor RB1 and STING. We uncovered that miR-181a mediated repression of RB1 initiated

tumor formation, caused profound DNA damage, and increased nuclear defects as well as GI. Normally, when p53 is already compromised these changes induce the STING pathway to cause cell death. However, we show that by simultaneously targeting STING miR-181a creates a cellular environment that is conducive to propagating these malignant cells. Our data suggests a unrecognized role for STING to prevent FTSEC transformation by detecting cytoplasmic dsDNA/GI, and that miR-181a inhibits this process. Importantly, miR-181a induction in patient tumors is associated with decreases in markers of tumor immunoreactivity suggesting that these tumors would be insensitive to immunotherapies. Taken together, our data show that miR-181a promotes FTSEC transformation through the combinatorial inhibition of a tumor suppressor gene and cell-intrinsic immunosurveillance signaling. Ultimately, these studies introduce a unique therapeutic drug target that can potentially re-sensitize tumors to immunotherapy.

## Results

**miR-181a promotes transformation of FTSECs in vitro.** In order to explore whether miR-181a could function as an oncomiR during the early stages of HGSOC development, we used three independent, patient-derived FTSEC cell lines (FT237, FT240, and FT246) immortalized (but not transformed) with genetic alterations found in HGSOC precursor lesions rather than viral oncoproteins. We then stably overexpressed mature miR-181a (pmiR-181a) to levels typically seen in HGSOC patient tumors (Fig. 1b and Supplementary Fig. 1B) or a scramble control with no known mammalian mRNA targets (pscram-miR). In the FT237 model we also stably overexpressed a miR-181a specific antagomiR (pmiR-181a + antimiR) to knockdown miR-181a expression and see which phenotype aspects were reversible. An initial indicator of increased transformation in the FT pmiR-181a vs pscram-miR cells was the loss of 2D contact inhibition (Fig. 1a), as well as increases in cell viability (Fig. 1c), clonogenicity (Fig. 1d), and formation of anchorage independent colonies (Fig. 1e, f). The anchorage independent growth results were particularly indicative of the transformation potential of miR-181a as only two stand-alone oncogenic alterations (CCNE1 amplification or YAP activation) have shown these effects in FTSECs[17,18]. Cell cycle analysis also showed an increase in the G2/M subpopulation of the FT pmiR-181a cells vs pscram-miR (Fig. 1g) which was consistent with the increases in proliferation seen in the pmiR-181a cells. MiR-181a overexpression also generated an approximately twofold to threefold increase in the >4 N subpopulation of cells (Fig. 1g), suggesting that the pmiR-181a cells had an increased frequency of large scale chromosomal aberrations (i.e., aneuploidy) and possibly genomic instability. Importantly, all transformation phenotypes observed were reversed in the pmiR-181a + antimiR cells (Supplementary Fig. 1C–H). These results show that miR-181a can act as a multifunctional oncomiR that promotes numerous aspects of FTSEC transformation.

**miR-181a overexpressing FTSECs form tumors in vivo.** We next investigated the in vivo tumorigenic capacity of miR-181a overexpressing FTSECs. Typically, in models of HGSOC transformation, at least two additional oncogenic "hits" are required for in vivo tumor formation. Subcutaneous cell injections of the pmiR-181a cells resulted in tumor formation in nine out of ten injected sites (Fig. 2a–d). In contrast, none of the pscram-miR and only one of the pmiR-181a + antimiR injection sites formed tumors (Fig. 2a–d). Growth kinetics of the pmiR-181a tumors were consistent with our in vitro data where the primary phenotype was the increase in anchorage independent growth rather

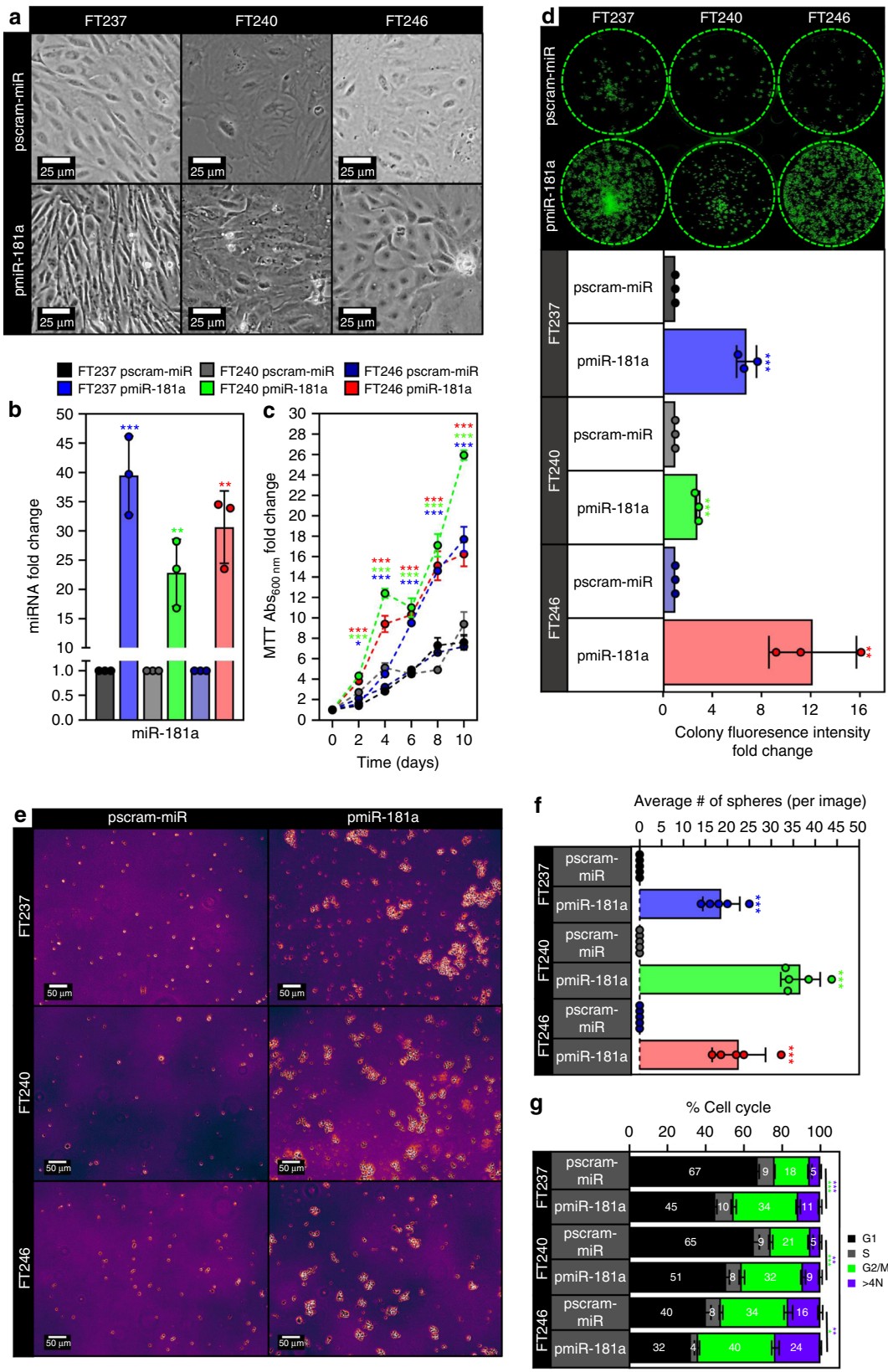

than proliferation (Fig. 2c). Histologically, the pmiR-181a tumors were consistent with HGSOC patient tumors (Fig. 2e). The tumors stained positive for the mullerian marker PAX8, the proliferation marker Ki-67, and γH2AX (a DNA damage marker

positively correlated with early transformation and genomic instability) (Fig. 2e). Next, we performed orthotopic cell injection in order to mimic HGSOC dissemination into the peritoneum, and only mice injected with pmiR-181a cells formed IP nodules

**Fig. 1 miR-181a promotes transformation of FTSECs in vitro. a** Phase contrast micrographs showing loss of contact inhibition in the FT pmiR-181a vs pscram-miR cells. All cells were plated at the same time at equal density and allowed to grow for 10 days. **b** Graph showing miR-181a expression levels for the FT pmiR-181a vs pscram-miR cell lines. **c** Graph showing increases in cell viability over a 10-day period for the FT pmiR-181a vs pscram-miR cells. Significance values are color coded to match the corresponding FT pmiR-181a cell line. **d** Colony formation assay showing increased survival and colony formation for the FT pmiR-181a vs pscram-miR cells with quantification (below). Colonies were stained with CellTag 700 at 10 days. Dashed green lines denote the culture plate well boundaries. **e** Micrographs showing increased anchorage independent growth of FT pmiR-181a vs pscram-miR cells. **f** Quantification of anchorage independent growth of FT pmiR-181a vs pscram-miR cells. Data represent $N = 5$. **g** Bar graph of the %Cell Cycle populations for the FT pscram-miR and FT pmiR-181a cells. All data are representative of $N = 3$ independent experiments unless otherwise stated. The measure of center for the error bars is given as the mean value unless otherwise stated. The statistical test used for data analysis is the two-sided Student's $t$ test unless otherwise stated. Error bars indicate ± standard deviation unless otherwise stated. $*p < 0.05$, $**p < 0.005$, $***p < 0.0005$.

and ascites (Fig. 2f–h). The IP nodules formed in the pmiR-181a mice recapitulated the spread of HGSOC in patients with IP nodules forming on the uterine serosa, intestines and peritoneum (Fig. 2f).

**miR-181a causes nuclear and mitotic defects in FTSECs**. Given the transformative ability of miR-181a, we next explored the mechanistic effects of this oncomiR on key drivers of cellular transformation. Based on the observed aneuploidy phenotype in the FT pmiR-181a cells, we investigated whether these cells displayed nuclear or mitotic defects which are characteristic features of HGSOC development linked to increased GI. Using fluorescence microscopy, it was apparent that the pmiR-181a cells had a higher frequency of structural nuclear defects such as multi-lobation, micronuclei, binucleation, trinucleation, and poly-nucleation (Fig. 3a and Supplementary Fig. 2A). We quantified the defects in nuclear shape by measuring the nuclei circularity. Normal epithelial cell nuclei have a circularity between 0.8 and 1.0 as was seen in our FT pscram-miR cells (Fig. 3a and Supplementary Fig. 2B). With miR-181a overexpression there was an approximate threefold to fourfold increase in cells with abnormal nuclei (Fig. 3a and Supplementary Fig. 2B). To further characterize nuclear shape defects and their effect on cell fate we performed live cell imaging using an SV40-GFP reporter to track nuclear shape before and after mitotic division. We found that the majority of pscram-miR cells maintained a normal nucleus in the parent and daughter cells (Supplementary Figs. 2C and 3A and Supplementary Movie 1A). In contrast, the pmiR-181a cells had abnormal parent cell nuclei giving rise to at least one daughter cell with an abnormal nucleus (Supplementary Figs. 2C and 3A and Supplementary Movie 1B). Overexpression of the miR-181a antagomiR in the pmiR-181a cells shifted the distribution back to the normal nuclear morphology and reversed the circularity defects (Fig. 3a, Supplementary Figs. 2C and 3A, and Supplementary Movie 1C).

We also noticed a high frequency of nuclear rupture events in the pmiR-181a cells. Abnormal nuclear morphology is often accompanied by transient ruptures in the nuclear membrane which can contribute to GI. We characterized the frequency of nuclear rupture events using two methods. The first was a static method looking at cytoplasmic promyelocytic leukemia protein (PML) aggregates. PML aggregates are structures that normally reside in the nucleus and are too large to pass through nuclear pores[19]. We observed a threefold to fourfold increase in the number of cytoplasmic PML + cells in the FT pmiR-181a cells (Supplementary Fig. 3B). Real-time monitoring using the SV40-GFP reporter showed 2% of pscram-miR cells experienced a nuclear rupture event, whereas 50% of the pmiR-181a cells had at least one nuclear rupture (Fig. 3b–d and Supplementary Movies 2A–D). We also observed an increase in cell division time for the pmiR-181a cells (Fig. 3e) which lead us to suspect increased frequency of mitotic and cytokinetic (MITOC) defects.

A variety of MITOC defects were increased at varying frequencies in the pmiR-181a including nucleoplasmic bridges, lagging chromosomes, failed cytokinesis, and multipolar cytokinesis (Fig. 3f, g and Supplementary Movies 3A–H). The increases in the MITOC defects observed in the pmiR-181a cells were reversible with addition of the miR-181a antagomiR (Fig. 3g).

Given the increased frequency of nuclear and MITOC defects seen in the pmiR-181a cells, we tracked the fate of these cells using an H2B-GFP reporter. The pmiR-181a cells had a majority of parent cells with abnormal multilobed nuclei and a higher subpopulation of multinucleate cells as compared with the pscram-miR cells (Supplementary Fig. 4A). This shift in parent nuclei status and higher incidence of MITOC abnormalities was reversible with the addition of the miR-181a antagomiR (Fig. 3g. Supplementary Fig. 4A). Analysis of daughter cell fate indicated that pscram-miR cells gave rise to a majority of normal daughter cells that either survived or underwent apoptosis (7% of pscram-miR daughter cells with abnormal fate outcomes) (Fig. 3g). In contrast, the pmiR-181a cells gave rise to a majority of abnormal daughter cells (92% of pmiR-181a daughter cells with abnormal fate outcomes) that survived up to 72 h, with increases in the number of multinucleate cells and cells undergoing multipolar cytokinesis (Fig. 3g). The p181a-antimiR cells had a daughter cell distribution comparable to control cells. The pmiR-181a cells had a dramatic increase in the population of abnormal daughter cells (67%) that survive vs either pscram-miR or antimiR cells (0%) (Fig. 3g). Of particular interest, there were increased numbers of pmiR-181a + antimiR cells undergoing cell death during mitosis or cytokinesis (Fig. 3g and Supplementary Fig. 4B, C). These data suggest that miR-181a overexpression promotes cell survival in the presence of mitotic abnormalities and when inhibited these cells underwent cell death. Taken together, these data show that miR-181a overexpression promotes as well as propagates nuclear and MITOC defects all of which can contribute to GI and early HGSOC tumor development.

**miR-181a promotes genomic instability in FTSECs**. Due to the increased nuclear and MITOC defects in pmiR-181a cells, we hypothesized that these cells would have an increased frequency of large scale GI. Gene set enrichment analysis (GSEA) in the FT237 models identified gene signatures associated with DNA damage and genomic instability as being upregulated in the pmiR-181a cells (Fig. 4a and Supplementary Fig. 5A–C). The addition of the miR-181a antagomiR reversed the changes in gene expression caused by miR-181a (Fig. 4a and Supplementary Fig. 5A–C). We next confirmed that miR-181a mediated increases in DNA damage by examining γH2AX foci. There was a profound increase in γH2AX foci per cell in pmiR-181a cells compared with control cells (Fig. 4b, c). We also found that nuclei circularity was inversely correlated with the number of γH2AX foci per cell across all cell lines (Fig. 4c) suggesting an intriguing functional link between the miR-181a induced nuclear

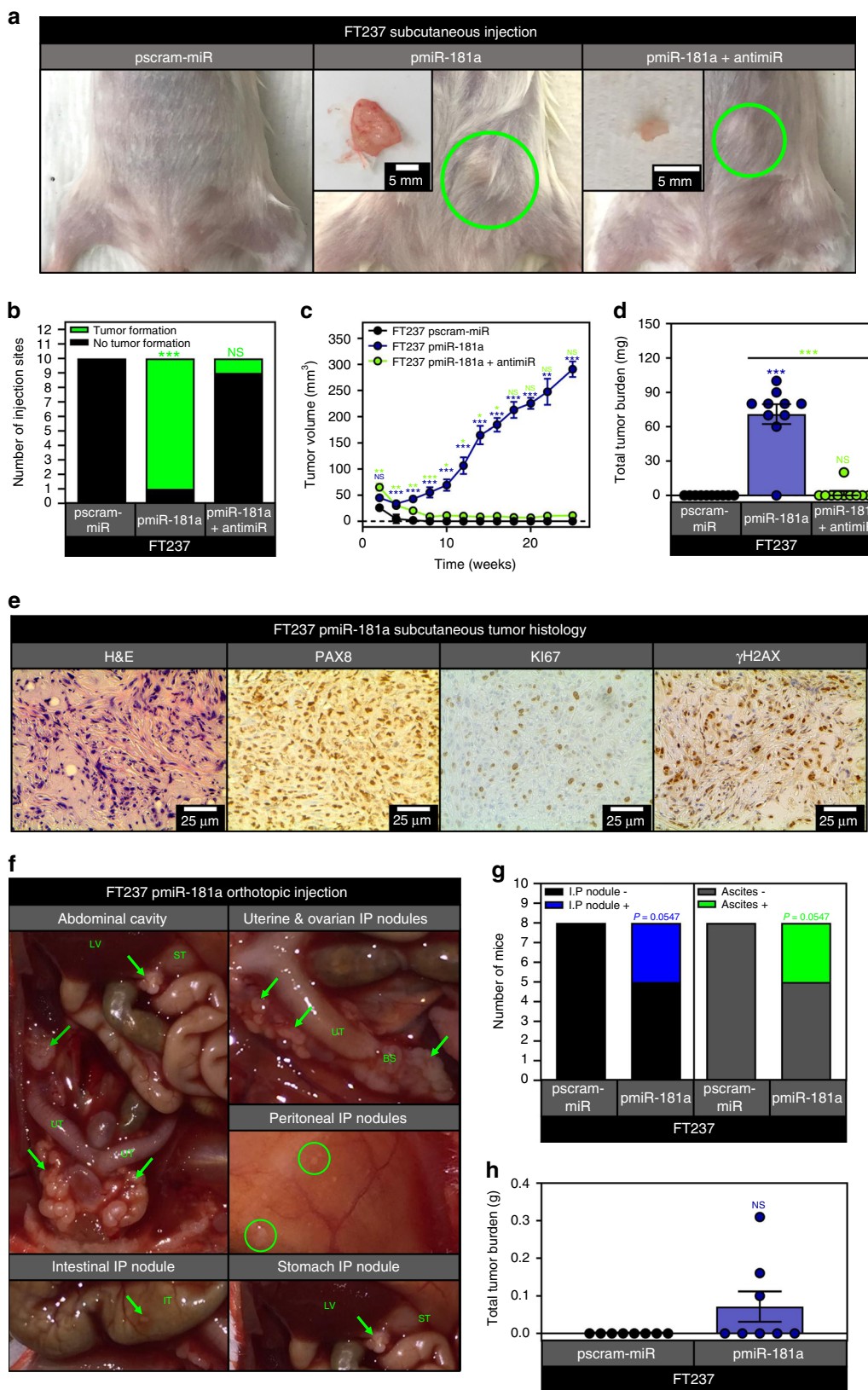

dysmorphia and DNA damage. In order to ascertain whether the increased DNA damage was an indicator of increased chromosomal and large scale GI we utilized a SNP array to characterize genome-wide gains/losses. Genomaps showed an increase in genomic instability in pmiR-181a cells that was reduced with miR-181a antagomiR addition (Fig. 4d). The fraction of the

genome altered increased from ~15% in the pscram-miR cells to 30% in the pmiR-181a cells (approaching what is typically seen HGSOC patient samples), and was reduced back to ~18% in the pmiR-181a + antimiR cells (Fig. 4e). Analysis of total copy number variation (CNV) events indicated roughly a threefold increase in CNV events in the pmiR-181a vs control cells (Fig. 4f).

**Fig. 2 miR-181a promotes FTSEC tumor formation in vivo. a** Pictures showing representative images of subcutaneous tumor formation at 25 weeks post injection for the FT237 pscram-miR, pmiR-181a, and pmiR-181a + antimiR groups with insets depicting excised tumors. **b** Graph showing the number of subcutaneous injection sites that formed palpable tumors by 25 weeks for each group of mice injected with either FT237 pscram-miR, pmiR-181a, or pmiR-181a + antimiR cells. **c** Graph depicting tumor growth kinetics for the FT237 pscram-miR, pmiR-181a, and pmiR-181a + antimiR subcutaneous injection groups. Error bars represent ±SEM. Significance values are color coded to match the corresponding mouse injection group. **d** Scatter plot showing total tumor burden for the FT237 pscram-miR, pmiR-181a, and pmiR-181a + antimiR subcutaneous injection groups. $N = 10$ for all groups. **e** Representative micrographs of subcutaneous FT237 pmiR-181a tumors stained with either H&E (far left), PAX8 (left), KI-67 (right), or γH2AX. **f** Representative images showing intraperitoneal nodule formation in FT237 pmiR-181a mice: (top left) gross anatomy (bottom left) IP nodule located on intestine (top right) magnified view of uterine and ovarian IP nodules, (middle right) peritoneal IP nodules, (bottom right) stomach IP nodule. **g** Graph showing the number of intraperitoneally injected mice that formed intraperitoneal nodules or ascites by 25 weeks for each group of mice injected with either FT237 pscram-miR or FT237 pmiR-181a cells. **h** Scatter plot showing total tumor burden for the FT237 pscram-miR and pmiR-181a intraperitoneal injection groups. $N = 8$ for both groups. All data are representative of $N = 3$ independent experiments unless otherwise stated. The measure of center for the error bars is given as the mean value unless otherwise stated. The statistical test used for data analysis is the two-sided Student's $t$ test unless otherwise stated. Fisher's exact test was used for statistical analysis in **b** and **g**. Mann–Whitney test was used for statistical analysis in **d** and **h**. Error bars indicate ± standard deviation unless otherwise stated. *$p < 0.05$, **$p < 0.005$, ***$p < 0.0005$.

Subtyping of the CNV events demonstrated a distinct pattern in the pmiR-181a cells with increases in LOH and duplication events (Fig. 4f). The disruptive effects of miR-181a overexpression on the genome were most apparent when analyzing the fraction of the genome with either LOH (FLOH) or amplification (FAMP). The FLOH increased sixfold and the FAMP increased fivefold in the pmiR-181a vs pscram-miR cells (Fig. 4g, h). Taken together, these data demonstrate that miR-181a overexpression in FTSECs promotes both increased DNA damage and GI, two defining features of HGSOC tumors.

**miR-181a promotes oncogenic changes in gene expression**. We next sought to determine the target genes driving the miR-181a transformation phenotypes. Global mRNA transcriptome analysis revealed 601 differentially expressed genes in the pmiR-181a cells compared with pscram-miR cells (Fig. 5a). In the pmiR-181a + antimiR cells 457 genes were differentially regulated when compared with pscram-miR cells (Fig. 5a). Using Ingenuity Pathway Analysis (IPA), we found that Cancer functions were the top ranked group in the Diseases & functions category (Fig. 5b) indicating that miR-181a overexpression promoted gene expression changes associated with oncogenic transformation. Interestingly, we found that the largest subpopulation of the altered cancer signatures in the pmiR-181a cells were ovarian cancer gene sets (Fig. 5c). In addition, a number of processes involved in HGSOC transformation had significant overlap with the pmiR-181a gene signature (Fig. 5d). The activation $z$-score for these cellular functions was reversed with addition of the miR-181a antagomiR (Fig. 5d) confirming that miR-181a drove the gene expression changes.

In order to identify the target(s) responsible for the observed miR-181a driven phenotypes, we first identified all the predicted miR-181a targets that were downregulated in the pmiR-181a cells. We next cross-referenced those predicted targets against the genes that were upregulated in the pmiR-181a + antagomiR cells. Through this analysis 70 candidate miR-181a targets were identified (Fig. 5e).

**miR-181a targets RB1 to promote FTSEC transformation**. Twenty-five out of the seventy candidates were associated with cellular functions related to the observed miR-181a phenotype in FTSECs. Of these, RB1 was one of the most promising candidates (Fig. 5e). The role of RB1 as a tumor suppressor in the early stages of HGSOC oncogenic transformation is well established. Gene sets associated with decreased RB1 function and increased G1/S transition were significantly enriched in the pmiR-181a cells and reversed in the pmiR-181a + antagomiR cells (Fig. 6a, b). We

next sought to validate RB1 as a direct target of miR-181a. We confirmed decreased RB1 mRNA and protein expression in the FT pmiR-181a vs pscram-miR cells (Fig. 6c, d and Supplementary Fig. 6A). Using an RB1 3′UTR luciferase assay, we confirmed direct targeting of miR-181a wherein we found that there was a ~30–40% decrease in the RB1 3′UTR luciferase activity in the FT pmiR-181a vs pscram-miR cells (Fig. 6e and Supplementary Fig. 6B). The miR-181a antagomiR rescued RB1 expression as well as RB1 3′UTR activity (Fig. 6e). In addition, there was no significant decrease in mutant RB1 3′UTR (RB1 MUT 3′UTR) reporter activity for any of the cell lines (Fig. 6e).

Next, to determine what aspects of the miR-181a phenotype were mediated by RB1 inhibition, we utilized stable shRNA knockdown of RB1 in the FT237 cells (Fig. 7a). The pmiR-181a and pshRB1 cells both showed comparable increases in proliferation, survival, and anchorage independent growth vs pscram-miR (Fig. 7b and Supplementary Fig. 7A, B). The pshRB1 cells had an increase in the G2/M subpopulation similar to the pmiR-181a cells, but a more modest increase in the >4 N subpopulation when compared with pscram-miR cells (Fig. 7c). FT237 pshRB1 cells also had in vivo tumor formation, growth kinetics, and tumor burden comparable to pmiR-181a cells (Fig. 7d, e). Taken together, these data indicated that RB1 knockdown in the FTSECs was able to phenocopy miR-181a's effects on cellular viability, anchorage independent growth, cellular survival, and tumor formation. Importantly, these phenotypes were driven specifically due to miR-181a targeting of RB1. To further confirm this we stably expressed miR-181a in FT194 cells (FTSECs immortalized with SV40 large T antigen, which functionally inactivates RB1) and found that it did not produce the transformation phenotypes seen in RB1-functional FTSECs (Supplementary Fig. 8A–E).

Though RB1 inhibition phenocopied the tumor initiating phenotypes observed in miR-181a overexpressing FTSECs, the nuclear defects and DNA damage phenotypes observed in the pshRB1 cells were consistently more diminished compared to the pmiR-181a cells (Supplementary Fig. 7C–F). In addition, examination of genomic instability in the pshRB1 cells showed a pronounced decrease in the total number of CNV events compared with the pmiR-181a cells (Fig. 7f and Supplementary Fig. 7G). The distribution of CNV events in the pshRB1 cells was also different from the pmiR-181a cells with the pshRB1 cells showing an increase in heterozygous losses, no difference in LOH events, and a diminished increase in amplification events compared with pmiR-181a cells (Fig. 7f). Interestingly, even though RB1 knockdown showed effects on nuclear circularity, DNA damage, and nuclear rupture, this was not sufficient to drive the same level or pattern of nuclear defects and genomic

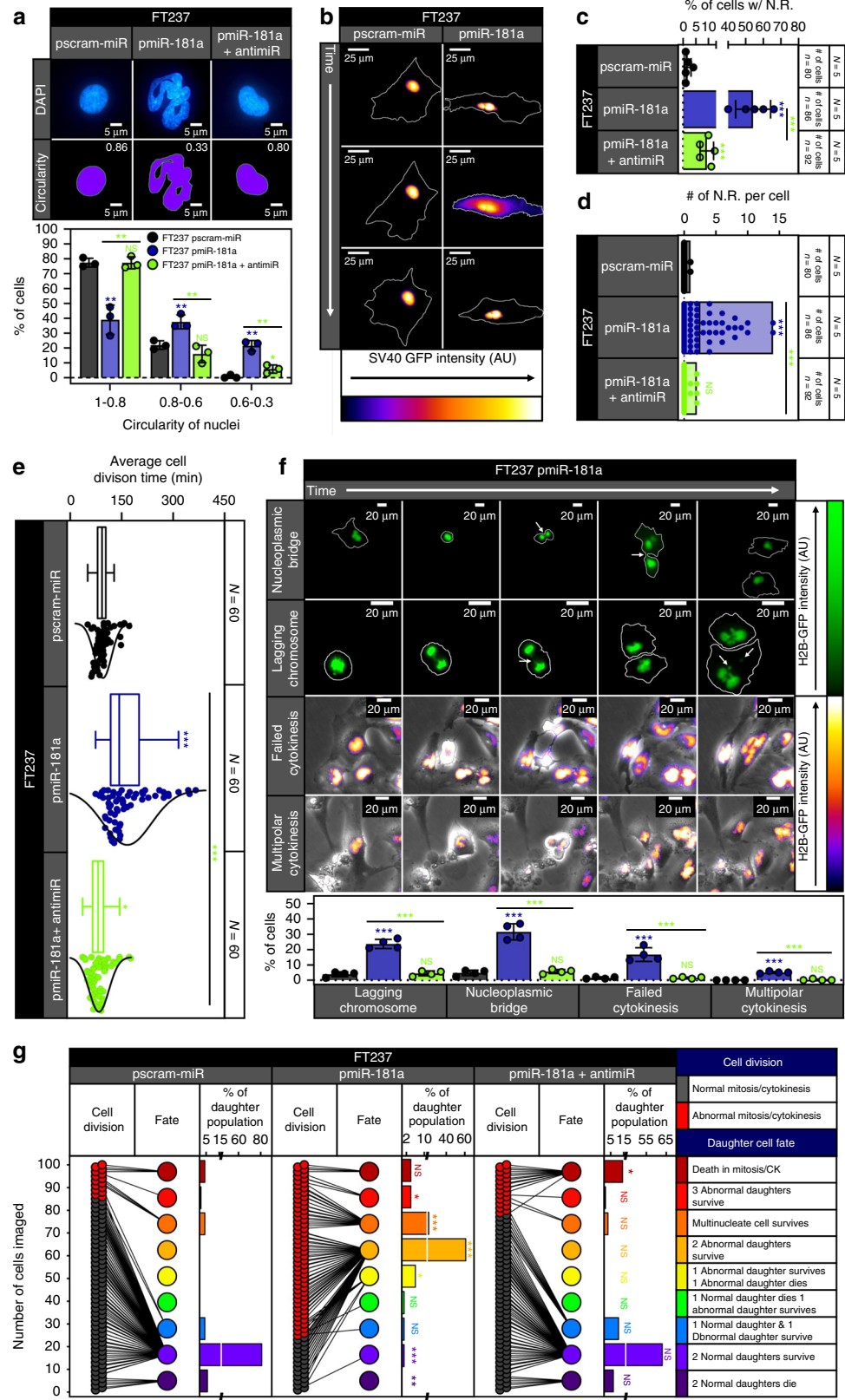

instability observed in the pmiR-181a cells. Thus, RB1 alone could not account for the full increase and propagation of cells with aneuploidy and nuclear defects observed in the pmiR-181a cells.

**miR-181a targets STING to bypass GI-triggered interferon response**. At this point, we had established a model whereby miR-181a promotes genomic instability and transformation in FTSECs in part by targeting the tumor suppressor RB1. However,

**Fig. 3 miR-181a increases nuclear rupture and mitotic/cytokinetic defects in FTSECs. a** Immunofluorescence micrographs of representative DAPI stained nuclei (top), matched circularity masks with circularity value displayed in upper right hand corner (middle), and graph showing the circularity distribution (bottom) from pscram-miR, pmiR-181a, and pmiR-181a + antimiR cells. **b** Time-lapse micrographs of pscram-miR cells (left) and a nuclear membrane rupture event in pmiR-181a cells (right) expressing SV40-GFP. White lines indicate the outline of the cell in each micrograph. **c** Quantification of the percent of cells with nuclear membrane rupture for the pscram-miR, pmiR-181a, and pmiR-181a + antimiR cells. $N = 5$ for each cell line, $\geq 80$ cells measured for each cell line. **d** Floating bar plot of the number of nuclear ruptures/cell for the pscram-miR, pmiR-181a, and pmiR-181a + antimiR cells. $N = 5$ for each cell line, $\geq 80$ cells measured for each cell line. **e** Box and whisker plot with distribution curves of the cell division times for the pscram-miR, pmiR-181a, and pmiR-181a + antimiR cells. $N = 3$ for each cell line, 60 cell divisions measured for each cell line. **f** Representative time-lapse micrographs of mitotic and cytokinetic defects in pmiR-181a cells expressing H2B-GFP with quantification below. White outlines in the NP bridge and Lag. Chrom. panels denote the outline of the cell in each image. $N = 4$ for each cell line. **g** Cell fate outcomes for pscram-miR, pmiR-181a, and pmiR-181a + antimiR cells with before and after plot (left) of cell division and daughter cell fate outcomes and percentage quantification of daughter cell fate (right). Color key for cell division and daughter fate outcomes are on right. $N = 100$ cell divisions were observed for each cell line. Chi-square analysis for statistical comparison between groups. All data are representative of $N = 3$ independent experiments unless otherwise stated. The measure of center for the error bars is given as the mean value unless otherwise stated. Two-sided Student's $t$ test was used unless otherwise stated. Error bars indicate ±standard deviation unless otherwise stated. N.R. nuclear rupture. $*p < 0.05$, $**p < 0.005$, $***p < 0.0005$.

one critical question that remained unanswered was how FTSECs overexpressing miR-181a were able to survive in the context of persistently high levels of GI and DNA damage. Our live cell imaging data supported the idea that the pscram-miR remained sensitive to GI induced by nuclear rupture or MITOC defects as these cells displayed a high propensity to undergo cell death when exposed to GI-inducing stimuli. In contrast, the pmiR-181a cells had a low propensity to undergo cell death when exposed to GI-inducing stimuli. P53 is generally regarded as the gate-keeper of genome stability as it is able to both maintain genome stability and induce cell death if the levels of GI-inducing stimuli become too great[20]. However, in our control FTSECs which were p53 deficient, only 10–15% of the genome exhibited GI (versus $\geq 60\%$ in transformed HGSOC). This suggested that other pathways are involved in maintaining the genomic integrity of FTSECs, and induce cell death when the levels of GI become too great.

Examination of our global mRNA expression data in the FTSECs revealed that interferon signaling was one of the top downregulated canonical signaling pathways in the pmiR-181a cells, and that this downregulation of interferon signaling was reversed with addition of miR-181a antagomiR (Supplementary Fig. 9A). Recent studies have shown that induction of GI and DNA damage can activate interferon signaling by stimulating the cytoplasmic DNA sensing cGAS-STING pathway leading to either cellular senescence or death[21,22]. Interestingly, STING was one of the 25 predicted miR-181a targets with transformation-related functions that we identified through RNA expression analysis (Fig. 5e). Given that STING has a conserved miR-181a binding site in its 3'UTR, we hypothesized that miR-181a simultaneously increases GI while inhibiting STING mediated cell death allowing GI high FT pmiR-181a cells to survive and expand. We found decreased STING mRNA and protein expression in the FT pmiR-181a vs pscram-miR cells (Fig. 8a, b and Supplementary Fig. 6C). In addition, a STING 3'UTR assay confirmed that miR-181a targeted STING (Fig. 8c and Supplementary Fig. 6D). The decreases in FT237 pmiR-181a cell mRNA/protein expression and 3'UTR activity were rescued with addition of miR-181a antagomiR (Fig. 8c and Supplementary Fig. 6D). We confirmed that miR-181a was targeting the STING 3'UTR in the FTSECs using a mutant STING 3'UTR (STING MUT 3'UTR) reporter construct with a mutated miR-181a binding site (Fig. 8c). Importantly, we observed that STING was not significantly downregulated in the FT237 pshRB1 cells (Fig. 8a). This indicated that the downregulation of STING in the FT237 pmiR-181a cells was a result of direct interaction of miR-181a and STING as opposed to a secondary effect of miR-181a targeting RB1.

Next, we sought to see if activating STING signaling using the STING agonist 2'3'-cGAMP would induce interferon signaling as well as cell death, and whether this would be inhibited in the presence of miR-181a. We found that cGAMP induced significant expression of interferon target genes that mediate interferon-induced cell death (IFIT2 and TNFSF10) as well as recruitment of immune cells (CXCL10) in the pscram-miR cells (Supplementary Fig. 9B) and this increase was significantly reduced in the pmiR-181a cells (Supplementary Fig. 9B). We also confirmed that cGAS was expressed in the FTSECs indicating the upstream signaling machinery necessary to activate STING was present (Supplementary Fig. 9D). We next queried the ability of cGAMP to induce cell death through intrinsic signaling mechanisms in the FTSECs. We found that pmiR-181a cells were less sensitive to cGAMP induced cell death compared with either pscram-miR, pmiR-181a + antimiR, or pshRB1 cells (Fig. 8d). To confirm that cGAMP was inducing cell death (rather than inhibiting proliferation), we used a GFP fluorescent reporter assay to measure cell death. We found that cGAMP treatment significantly increased the percentage of pscram-miR and pmiR-181a + antimiR GFP + dead cells. This cGAMP induced cell death was decreased in the pmiR-181a cells (Fig. 8e). These findings were consistent with the suppression of STING signaling activation in miR-181a expressing cells (Fig. 8a–c). We next sought to determine if miR-181a upregulation affected the extrinsic mechanisms of STING-mediated tumor suppression in FTSECs. STING signaling acts as a key barrier to oncogenic progression through both regulation of cell-intrinsic interferon-induced cell death and the recruitment/activation of the immunosurveillance machinery. Our mRNA expression data of interferon inducible genes indicated that CXCL10 (an important chemokine in STING mediated immune cell recruitment) was repressed in the pmiR-181a cells, and we validated these data using a CXCL10 ELISA (Supplementary Fig. 9C). We found that secreted CXCL10 was reduced by ~50% in the pmiR-181a cells (Supplementary Fig. 9C). In addition, secreted CXCL10 levels were rescued with the addition of miR-181a antagomiR, and were not decreased in the pshRB1 cells to the same magnitude as the pmiR-181a cells (Supplementary Fig. 9C).

We next wanted to confirm that the miR-181a driven bypass of GI-induced cell death was a direct consequence of on-target binding to STING. To investigate this, we generated FT237 cells with stable knockdown of STING using two different shRNA hairpins. We confirmed STING knockdown in the two FT237 shSTING cell lines (shST #1 or shST #2) (Fig. 9a). We then confirmed that the shSTING cells were resistant to STING induced cell death by treating the cells with either lipofectamine

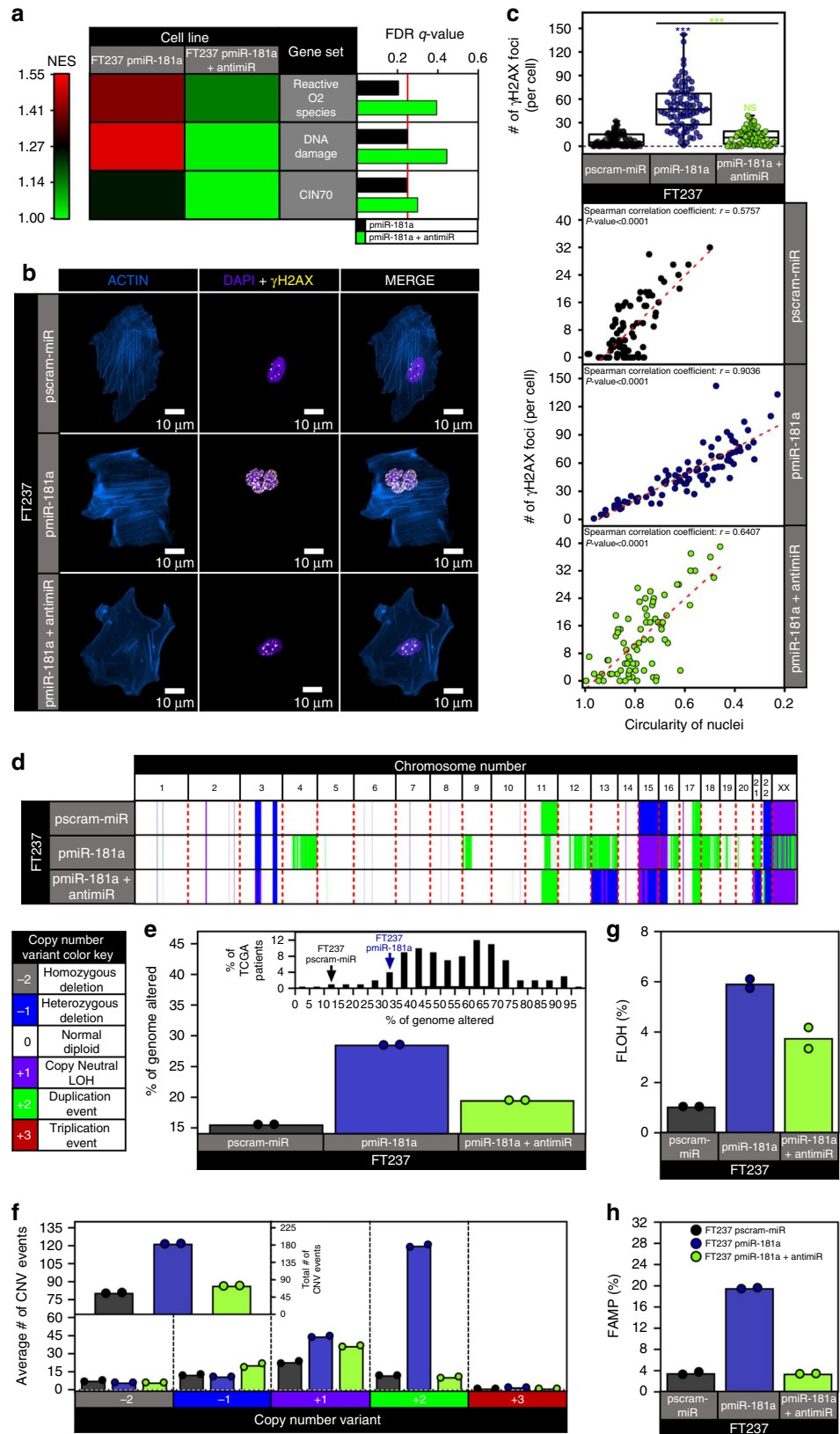

vehicle or cGAMP. (Fig. 9b). We then characterized the shSTING cells vs the pmiR-181a cells to assess the effects of STING knockdown on genomic stability, DNA damage, and transformation. Interestingly, knockdown of STING was sufficient to increase cell proliferation and clonogenic survival, consistent with the increases seen in the pmiR-181a cells (Fig. 9c, d). The

shSTING cells also had increased anchorage independent growth compared with the pscram-miR cells (Fig. 9e). In terms of cell cycle profile, STING knockdown produced similar shifts in cell cycle sub-populations as miR-181a overexpression including increases in the G2/M and aneuploid sub-populations (Fig. 9f, g). STING knockdown also increased the frequency of nuclear

**Fig. 4 miR-181a promotes DNA damage and drives genomic instability. a** GSEA of significantly altered gene sets associated with genomic instability in pmiR-181a cells. Heatmap (left) of normalized enrichment scores for either pmiR-181a vs pscram-miR or pmiR-181a + antimiR vs pscram-miR. Bar graph of FDR q-values for the pmiR-181a and pmiR-181a + antimiR GSEA results. The red line indicates the significance cut-off of 0.25. Multiple testing adjustments were made using FDR correction according to default GSEA parameters. **b** Representative immunofluorescence micrographs of γH2AX staining in pscram-miR, pmiR-181a, and pmiR-181a + antimiR cells. Purple indicates staining of actin with ActinRed, red indicates staining of nuclei with DAPI, green indicates staining of γH2AX foci. **c** Plot showing inverse correlation between the number of γH2AX foci for a cell nucleus and the corresponding circularity of the same cell nucleus. Spearman correlation coefficients and p values are displayed. **d** Genomap of copy number variants detected by SNP array in pscram-miR, pmiR-181a, and pmiR-181a + antimiR cells with color key below. **e** Graph depicting percent of the genome altered in pscram-miR, pmiR-181a, and pmiR-181a + antimiR cells. Inset graph shows the % genome altered of the FT cell lines in the context of % genome altered distribution for TCGA HGSOC patients. $N = 2$ for all cell lines. **f** Graph showing the number of events for each CNV subtype in pscram-miR, pmiR-181a, and pmiR-181a + antimiR cells. Inset graph shows the total number of CNV events for pscram-miR, pmiR-181a, and pmiR-181a + antimiR cells. $N = 2$ for all cell lines. **g** Fraction of the genome amplified in pscram-miR, pmiR-181a, and pmiR-181a + antimiR cells. $N = 2$ for all cell lines. **h** Fraction of the genome with loss of heterozygosity in pscram-miR, pmiR-181a, and pmiR-181a + antimiR cells. $N = 2$ for all cell lines. All data are representative of $N = 3$ independent experiments unless otherwise stated. The measure of center for the error bars is given as the mean value. Two-sided Student's t test was used unless otherwise stated. Error bars indicate ±standard deviation. *$p < 0.05$, **$p < 0.005$, ***$p < 0.0005$.

membrane ruptures and γH2AX protein expression to levels observed in pmiR-181a cells (Fig. 9h, i).

In addition to STING knockdown, we wanted to assess whether re-expression of STING could inhibit the miR-181a mediated transformation phenotypes and re-sensitize the pmiR-181a cells to STING induced cell death. We stably overexpressed STING lacking a 3′UTR (STING overexpression or STOE) in the pscram-miR (scram STOE) and pmiR-181a cells (m181 STOE) (Supplementary Fig. 10A). We confirmed that STING overexpression sensitized the pscram-miR cells to STING activation and re-sensitized the pmiR-181a cells to STING activation (Supplementary Fig. 10B). We then proceeded to investigate which miR-181a transformation phenotypes were inhibited by STING re-expression. We saw a decrease in the proliferation and anchorage independent growth in the pmiR-181a STOE vs pmiR-181a cells (Supplementary Fig. 10C, D). Cell cycle analysis indicated that STING overexpression in the pscram-miR cells produced an increase in the G2/M and >4N sub-populations as well as a fivefold increase of the Sub-G1 cell population (Supplementary Fig. 10E, F). These data suggested that STING overexpression caused pscram-miR cells with mitotic/cytokinetic defects to arrest and eventually undergo cell death. STING overexpression had a similar effect on the pmiR-181a cells increasing the G2/M and Sub-G1 cell populations but to a lesser degree (Supplementary Fig. 10E, F). This was most likely due to the fact that miR-181a was still able to target the endogenous pool of STING in the pmiR-181a STOE cells. We observed that STING overexpression was sufficient to decrease the percentage of cells with nuclear rupture in the pmiR-181a cells back to pscram-miR levels (Supplementary Fig. 10G). We also found that STING overexpression caused a decrease in the levels of γH2AX in both the pscram-miR and pmiR-181a cells (Supplementary Fig. 10H) most likely through the induction of cell death in these genomically unstable cells. Finally, it is well established that STING acts as a check against genomic instability and transformation not only by activating interferon signaling to induce cell death, but also by inducing senescence and promoting the senescence associated secretory phenotype to clear cells with GI. We therefore investigated whether miR-181a targeting of STING affected senescence in the presence or absence of STING activation. We found increased β-galactosidase (β-gal) positivity in pscram-miR cells when stimulated with cGAMP with no increases noted in the pmiR-181a cells (Supplementary Fig. 9E). Furthermore, STING overexpression in pscram-miR cells resulted in an increase in the baseline number of senescent cells further increasing upon cGAMP stimulation (Supplementary Fig. 9E). Interestingly, STING overexpression in the pmiR-181a cells did

not produce an increase in senescence regardless of STING activation. A likely explanation for this would be that miR-181a can still target the downstream effectors of STING induced senescence, which include the miR-181a target RB1. Collectively, these data indicate that FT pmiR-181a cells that harbor profound nuclear defects and nuclear rupture (resulting in cytoplasmic DNA increase and STING activation) are able to propagate and survive due to the direct inhibition of STING.

**miR-181a is inversely correlated with immune activation in HGSOC patient tumors.** Given the extensive functional impact of the miR-181a-STING signaling axis in early HGSOC transformation models, we next sought to determine the clinical relevance of miR-181a-STING signaling in relation to the patient intratumoral immune microenvironment. We found that STING mRNA expression inversely correlated with miR-181a expression in the TCGA-SOC patient population ($r = -0.2311$, p value < 0.0001, $N = 288$) (Fig. 10a), suggesting that miR-181a can target STING in advanced stage HGSOC tumors. We next examined whether miR-181a targeting of STING in HGSOC tumors could promote an immunosuppressive microenvironment. Interestingly, through the analysis of a dataset developed by Thorrson et al. we found that miR-181a expression inversely correlated with important metrics of immune activation within the TCGA-SOC patient tumors (Fig. 10b–e)[23]. Importantly, in agreement with our miR-181a-STING correlation data, Interferon Gamma Response was also decreased in the miR-181a High vs Low tumors ($r = -0.2858$, p value < 0.0001) (Fig. 10b). We also observed a general decrease in lymphocyte infiltration in the miR-181a High vs Low tumors as shown by the reduction in leukocyte fraction ($r = -0.1779$, p value = 0.0025), Lymphocyte Infiltration Signature ($r = -0.2745$, p value < 0.0001) and infiltrating M1 macrophages ($r = -0.1304$, p value = 0.0272) (Fig. 10d, e). Taken together these data demonstrate that patient tumors with high miR-181a expression have reduced STING expression and concomitant decrease in immune cell infiltration.

## Discussion

Intervention and treatment at the earliest possible stage of cancer development is imperative for favorable survival outcomes. In the absence of a robust knowledge of the mechanisms driving early HGSOC tumorigenesis, the prospects for developing effective early detection strategies to prevent metastatic progression remain dim. Progress has been made toward elucidating the natural history of HGSOC beginning with the identification of the FTSE as the primary site of origin for the majority of HGSOCs. Experimental models of HGSOC transformation have uncovered

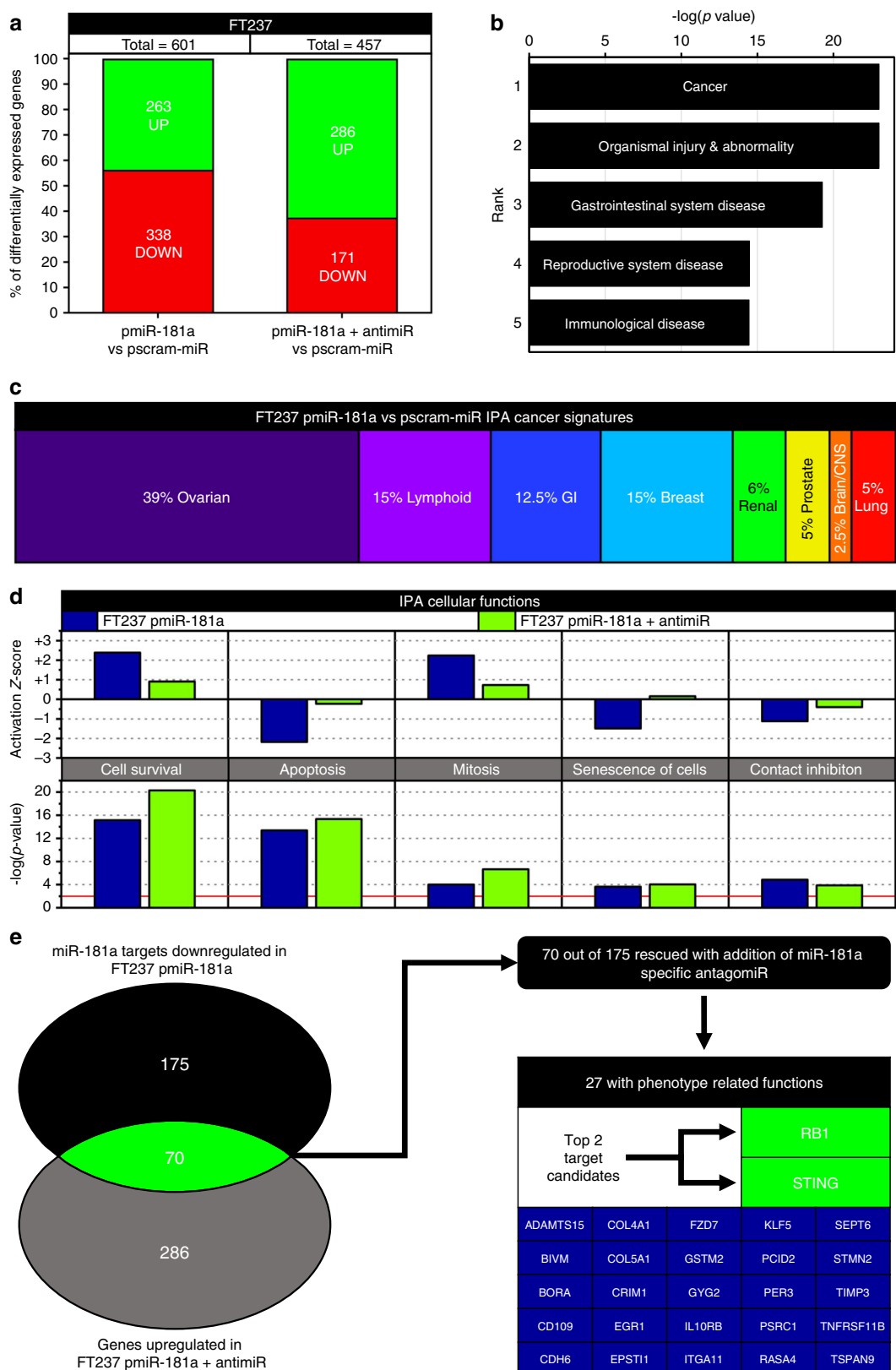

important driver genes such as TP53, BRCA1/2, CCNE1, and RB1.[24]. The main limitation is that these models do not easily translate into a useful preclinical model of early stage HGSOC transformation to allow for the development of effective early intervention strategies. The current state of the art is to demonstrate that these driver genes can phenotypically produce HGSOC but how they do it, particularly the mechanisms responsible for successful progression from normal tubal epithelium to p53 signature to STIC, remain incompletely defined.

**Fig. 5 miR-181a promotes tumorigenic changes in gene expression associated with oncogenic transformation and ovarian cancer. a** Graph of the percent of differentially expressed genes in FT237 pmiR-181a vs pscram-miR and FT237 pmiR-181a + antimiR vs pscram-miR along with the total number of genes differentially expressed for each cell line. **b** Graph of the –log(p values) for the top 5 ranked IPA Diseases and Functions groups significantly associated with the FT237 pmiR-181a cells. **c** Graph showing the relative percentages of IPA Cancer Signatures subgroups significantly associated with the FT237 pmiR-181a cells. **d** Graph comparing IPA Cellular Functions associated with tumorigenesis in the FT237 pmiR-181a vs pmiR-181a and antimiR cells. (Left) graph of IPA Cellular Functions Activation z-scores for the FT237 pmiR-181a and pmiR-181a + antimiR. (Bottom) graph of IPA Cellular Functions –log (p values) for the FT237 pmiR-181a and pmiR-181a + antimiR. **e** Diagram of the criterion filter selection process used to determine the miR-181a targets driving transformation and genomic instability in the FTSECs. All data are representative of N = 3 independent experiments unless otherwise stated. The measure of center for the error bars is given as the mean value unless otherwise stated. The statistical test used for data analysis is the two-sided Student's t test unless otherwise stated. Error bars indicate ±standard deviation unless otherwise stated. *p < 0.05, **p < 0.005, ***p < 0.0005.

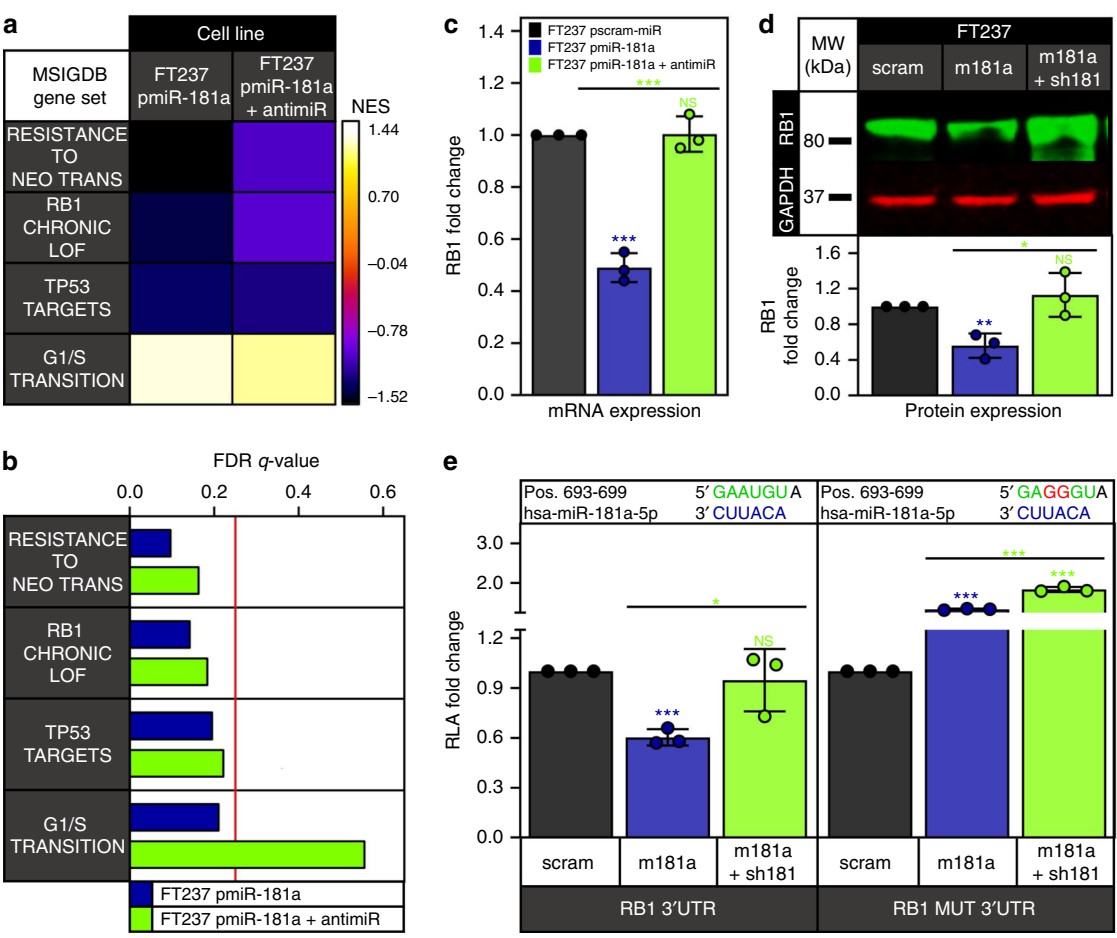

**Fig. 6 miR-181a targets RB1 to promote oncogenic transformation of FTSECs. a** Heatmap of normalized enrichment scores for either FT237 pmiR-181a vs pscram-miR or FT237 pmiR-181a + antimiR vs pscram-miR GSEA of significantly altered gene sets associated with RB1 knockdown and oncogenic transformation. **b** Bar graph of FDR q-values for the FT237 pmiR-181a and pmiR-181a + antimiR GSEA results. The red line indicates the significance cut-off of 0.25. Multiple testing adjustments were made using FDR correction according to default GSEA parameters. **c** Graph showing RB1 mRNA expression in the FT237 pscram-miR, pmiR-181a, and pmiR-181a + antimiR cells. **d** Representative western blot of RB1 expression levels in FT237 pscram-miR, pmiR-181a, and pmiR-181a + antimiR cells with quantification below. **e** Graph showing RB1 3'UTR and mutant 3'UTR relative luciferase activity for the FT237 pscram-miR, pmiR-181a, and pmiR-181a + antimiR cells. miR-181a binding sites within the 3'UTR are depicted in green. Mutated nucleotides within the 3'UTR are depicted in red. All data are representative of N = 3 independent experiments unless otherwise stated. The measure of center for the error bars is given as the mean value unless otherwise stated. The statistical test used for data analysis is the two-sided Student's t test unless otherwise stated. Error bars indicate ± standard deviation unless otherwise stated. *p < 0.05, **p < 0.005, ***p < 0.0005. Full western blots are shown in Supplementary Fig. 11.

One of the hallmark characteristics of HGSOC and its precursor lesions is large scale genomic instability (LSGI) characterized by frequent gains and losses of chromosomal regions and whole chromosomes. LSGI is one of the key initiating events in oncogenic transformation that then persists throughout the tumorigenic process[25–28]. Current tumorigenesis models are based on a gradualist theory of cancer cell evolution where driver mutations are accumulated slowly over time based on various

selective pressures. The primary limitation of this theory is that it does not adequately explain the evolutionary genomic patterns and the resultant phenotypes observed in HGSOC. Increasing evidence is emerging that punctuated LSGI is an initiating event in transformation and such punctuation events can serve as an inflection point that drive the earliest malignant transitions in HGSOC. Logically, it would follow that there are molecular/microenvironmental drivers that instigate punctuated LSGI in

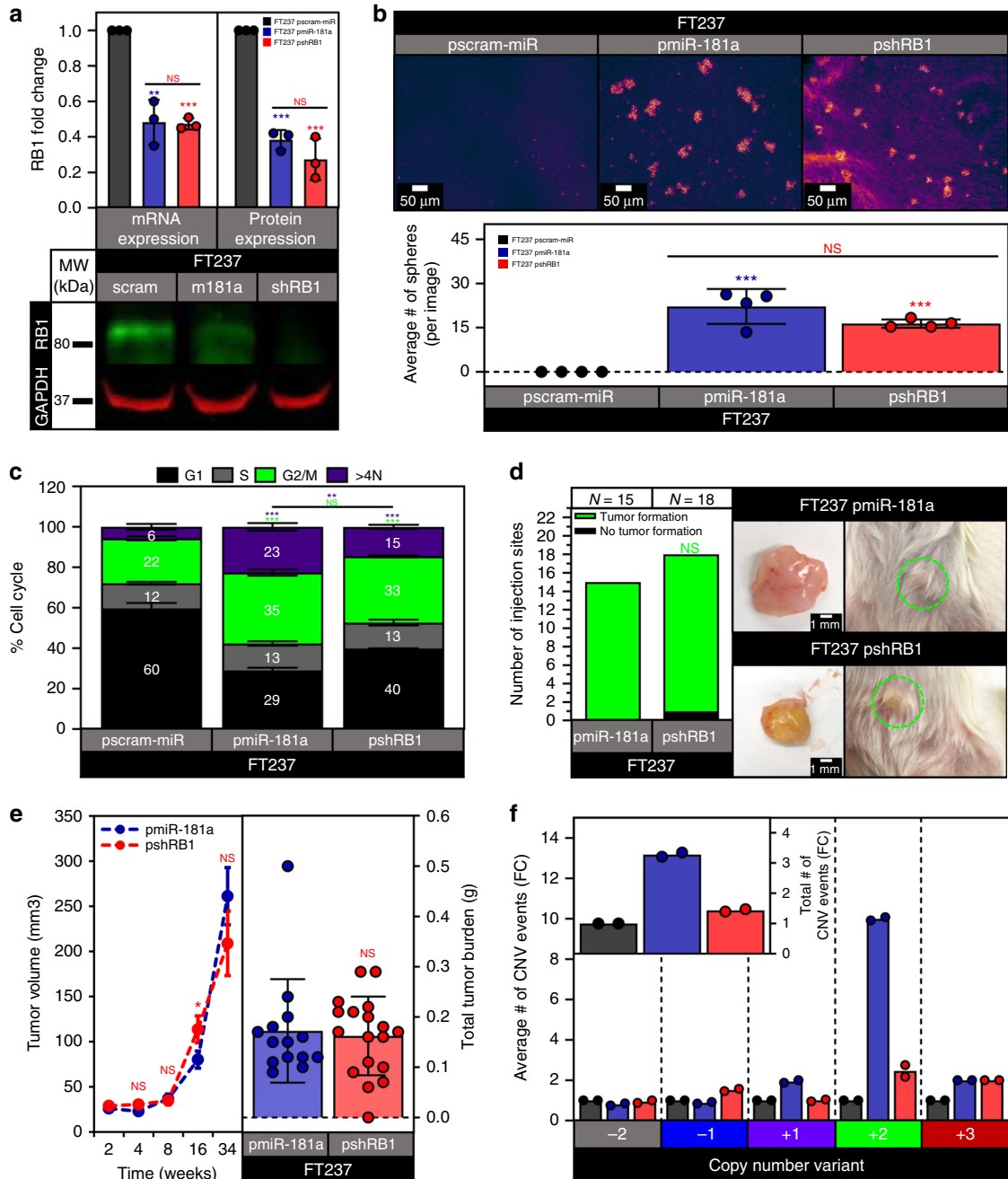

**Fig. 7 Knockdown of RB1 phenocopies miR-181a mediated transformation. a** Graphs showing RB1 mRNA and protein expression in the FT237 pscram-miR, pmiR-181a, and shRB1 cells (top) and a representative western blot of RB1 expression levels in FT237 pscram-miR, pmiR-181a, and shRB1 cells (bottom). **b** Micrographs showing anchorage independent growth of FT237 pscram-miR, pmiR-181a, and pshRB1 cells with quantification below. **c** Bar graph of the % Cell Cycle populations for the FT237 pscram-miR, pmiR-181a, and pshRB1 cells. **d** Pictures showing representative images of subcutaneous tumor formation at 25 weeks post injection for the FT237 pmiR-181a, and pshRB1 groups with insets depicting excised tumors (right) and graph of the number of injection sites that formed palpable tumors (left). Fisher's exact test was used for statistical analysis. **e** Graph depicting tumor growth kinetics for the FT237 pmiR-181a and pshRB1 subcutaneous injection groups (left). Error bars represent ±SEM. Significance values are color coded to match the corresponding mouse injection group. Scatter plot showing total tumor burden for the FT237 pmiR-181a, and pshRB1 subcutaneous injection groups (right). $N = 15$ for the FT237 pmiR-181a group, $N = 18$ for the FT237 pshRB1 group. Mann–Whitney test was used for statistical analysis. **f** Graph showing the fold change of total number of CNV events for FT237 pscram-miR, pmiR-181a, and pshRB1 cells (inset) and the fold change of the number of events for each CNV subtype in FT237 pscram-miR, pmiR-181a, and pshRB1 cells. $N = 2$ for all cell lines. All data are representative of $N = 3$ independent experiments unless otherwise stated. The measure of center for the error bars is given as the mean value unless otherwise stated. The statistical test used for data analysis is the two-sided Student's $t$ test unless otherwise stated. Error bars indicate ± standard deviation unless otherwise stated. *$p < 0.05$, **$p < 0.005$, ***$p < 0.0005$. Full western blots shown in Supplementary Fig. 11.

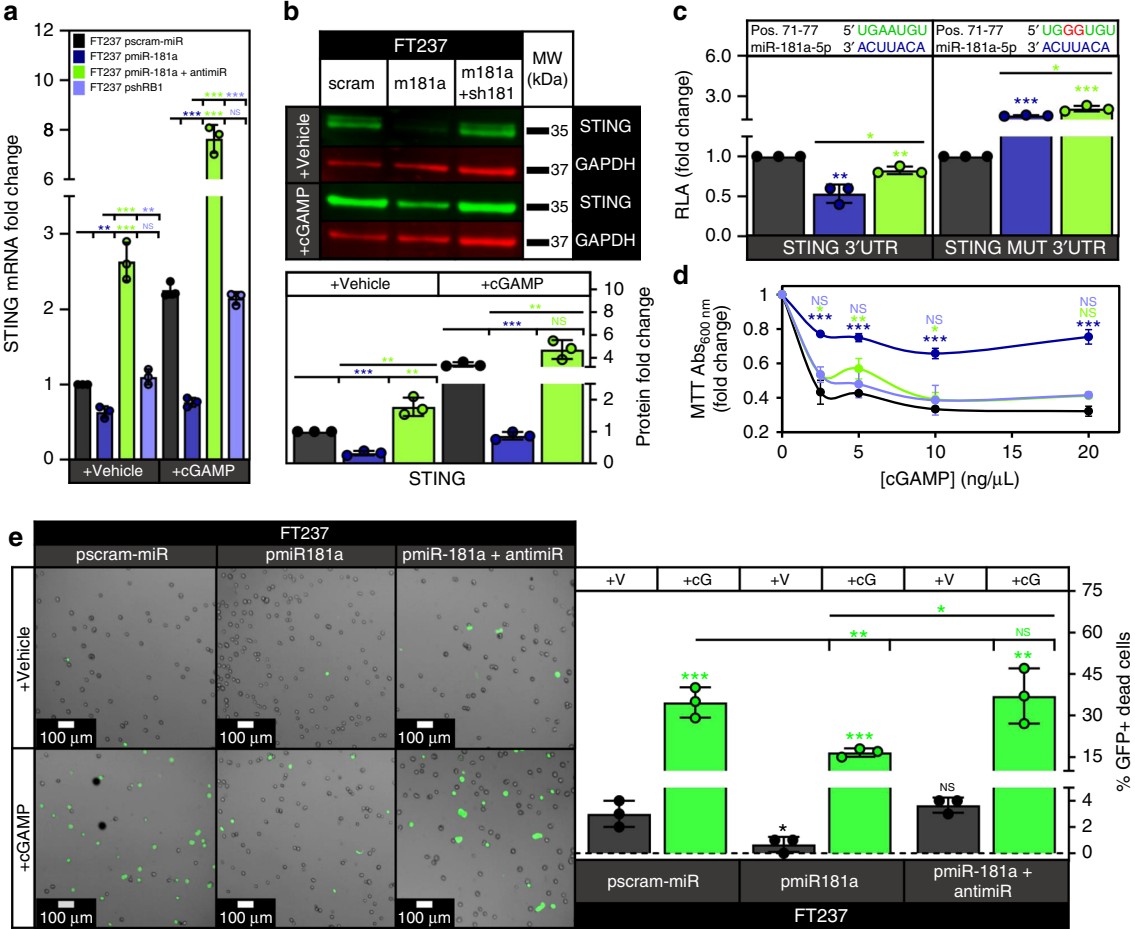

**Fig. 8 miR-181a allows FTSECs to bypass intrinsic interferon response by targeting STING. a** Graph showing STING mRNA expression in the FT237 pscram-miR, pmiR-181a, pmiR-181a + antimiR, and pshRB1 cells treated with either lipofectamine vehicle or the STING agonist cGAMP. **b** Representative western blot showing STING protein expression in the FT237 pscram-miR, pmiR-181a, and pmiR-181a + antimiR cells treated with either lipofectamine or cGAMP with quantification below. **c** Graph showing STING 3′UTR and mutant 3′UTR relative luciferase activity for the FT237 pscram-miR, pmiR-181a, and pmiR-181a + antimiR cells. miR-181a binding sites within the 3′UTR are depicted in green. Mutated nucleotides within the 3′UTR are depicted in red. **d** Graph showing cell viability of FT237 pscram-miR, pmiR-181a, pmiR-181a + antimiR, and pshRB1 cells 24 h after treatment with either lipofectamine vehicle or lipofectamine + increasing doses of cGAMP. **e** Images showing representative micrographs (left) and quantification (right) of FT237 pscram-miR, pmiR-181a, and pmiR-181a + antimiR GFP + dead cells 24 h after treatment with either lipofectamine vehicle or lipofectamine + 10 μg of cGAMP. All data are representative of N = 3 independent experiments unless otherwise stated. The measure of center for the error bars is given as the mean value unless otherwise stated. The statistical test used for data analysis is the two-sided Student's *t* test unless otherwise stated. Error bars indicate ± standard deviation unless otherwise stated. *$p < 0.05$, **$p < 0.005$, ***$p < 0.0005$. Full western blots shown in Supplementary Fig. 11.

FTSECs, and these drivers would be critical in elucidating the initiating events of HGSOC as well as developing early intervention strategies. In this study, we demonstrate that increased miR-181a expression promotes FTSEC transformation through the initiation of punctuated LSGI while simultaneously conferring survival advantages (by suppressing the innate immune response to nucleic acid sensing) permitting the increased LSGI to persist and expand.

We decided to investigate miR-181a as an oncomiR in HGSOC tumorigenesis given that previous work demonstrated that miR-181a could act as a driver of HGSOC tumor initiation in the cancer stem cell setting as well as a metastamiR in fully transformed HGSOC. In addition, high miR-181a expression correlates with poor outcomes and is located in a frequently amplified region of the genome in HGSOC[29]. Despite the fact that models of early HGSOC transformation have become more prevalent in the past 5–10 years there have been relatively few investigations of early HGSOC transformation drivers. This is even more reflective of miRNA drivers with only one miRNA, miR-182, being reported as

a potential oncomiR in early HGSOC development[30,31]. Our initial characterization of the miR-181a-transduced cells in vitro showed a striking increase in transformation phenotypes (Fig. 1). Of particular interest was the fact that the FT pmiR-181a cells showed an increase in the >4N population of cells. This indicated not only did miR-181a drive an increase in transformation, but also increased aneuploidy and LSGI making it a distinctive oncomiR.

We found that miR-181a alone was sufficient to promote subcutaneous and intraperitoneal orthotopic tumor growth, while the targeted inhibition of miR-181a abrogated this tumor formation (Fig. 2). The majority of FTSEC transformation studies have shown that only MYC or HRAS activation is necessary following hTERT immortalization and disruption of p53/RB1 signaling to produce HGSOC[32,33]. Others have shown that both MYC and HRAS signaling are required along with hTERT immortalization and disruption of p53/RB1 and PP2A via SV40 Large and Small T antigens[34,35]. With the exception of Karst et. al, all of these studies have made use of viral oncoproteins (and in

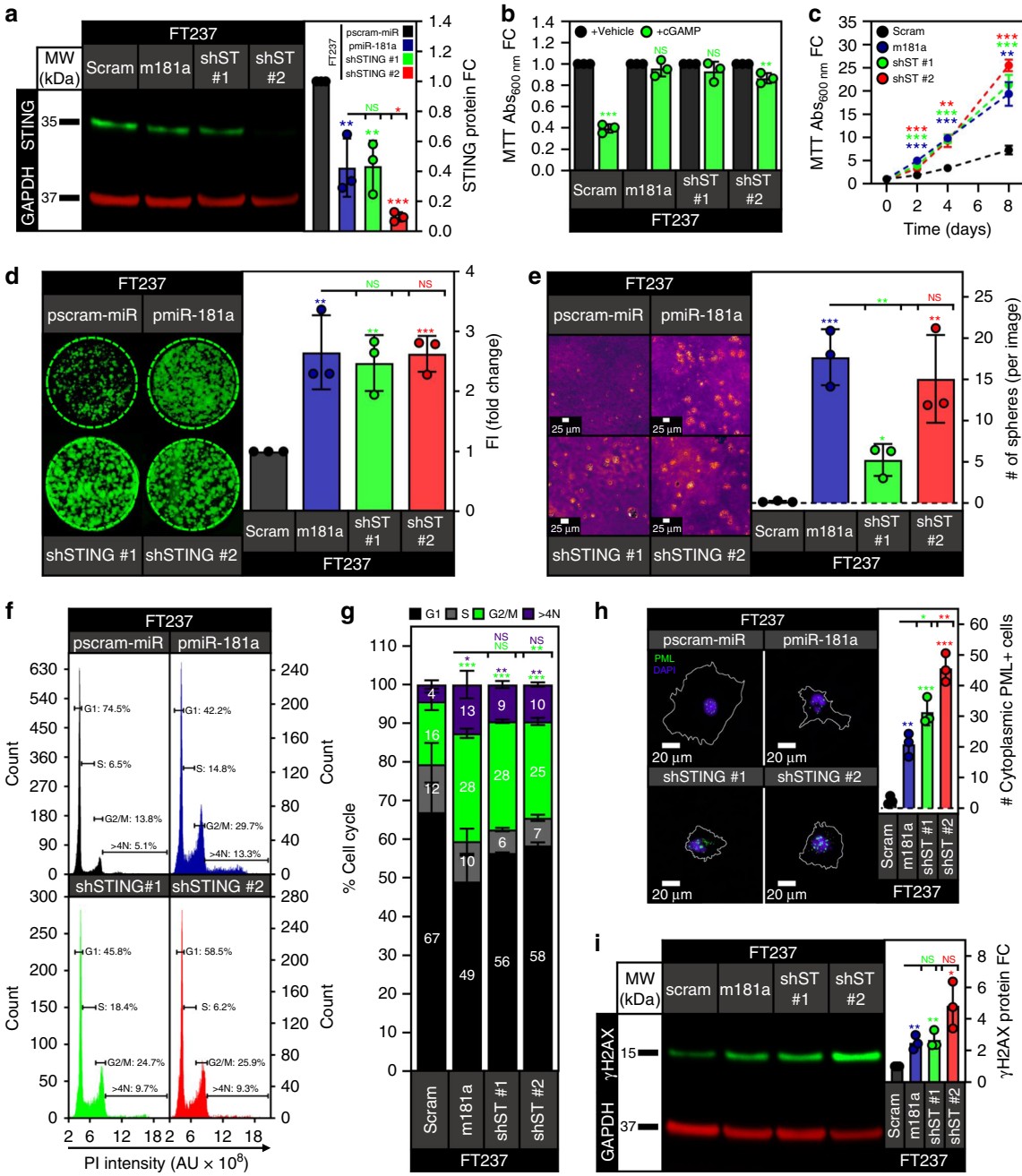

**Fig. 9 Knockdown of STING phenocopies miR-181a overexpression in FTSECs. a** Representative western blot of STING protein expression in FT237 pscram-miR, pmiR-181a, shSTING #1, and shSTING #2 cells with quantification on the right. **b** Graph of cell viability for FT237 pscram-miR, pmiR-181a, shSTING #1, and shSTING #2 cells 24 h after treatment with either lipofectamine vehicle or lipofectamine + 10 μg of cGAMP (top). **c** Graph of cell viability growth curves for FT237 pscram-miR, pmiR-181a, shSTING #1, and shSTING #2 cells (bottom). **d** Colony formation assay showing survival and colony formation for the FT237 pscram-miR, pmiR-181a, shSTING #1, and shSTING #2 cells with quantification on the right. Colonies were stained with CellTag 700 at 10 days. Dashed green lines denote the culture plate well boundaries. **e** Micrographs showing anchorage independent growth of FT237 pscram-miR, pmiR-181a, shSTING #1, and shSTING #2 cells with quantification on the right. **f** Representative graphs of cell cycle profiles for FT237 pscram-miR, pmiR-181a, shSTING #1, and shSTING #2 cells. **g** Graph depicting quantification of cell cycle subpopulations for FT237 pscram-miR, pmiR-181a, shSTING #1, and shSTING #2 cells. **h** Immunofluorescence micrographs of representative PML staining for the FT237 pscram-miR, pmiR-181a, shSTING #1, and shSTING #2 cells. PML bodies are stained green, DAPI stained nuclei are colored purple, and the outline of the cell is depicted in white. Quantification of the % cytoplasmic PML + cells for each cell line is located on the right. **i** Representative western blot of γH2AX protein levels in the FT237 pscram-miR, pmiR-181a, shSTING #1, and shSTING #2 cells with quantification on the right. All data are representative of $N = 3$ independent experiments unless otherwise stated. The measure of center for the error bars is given as the mean value unless otherwise stated. The statistical test used for data analysis is the two-sided Student's $t$ test unless otherwise stated. Error bars indicate ± standard deviation unless otherwise stated. $*p < 0.05$, $**p < 0.005$, $***p < 0.0005$. Full western blots shown in Supplementary Fig. 11.

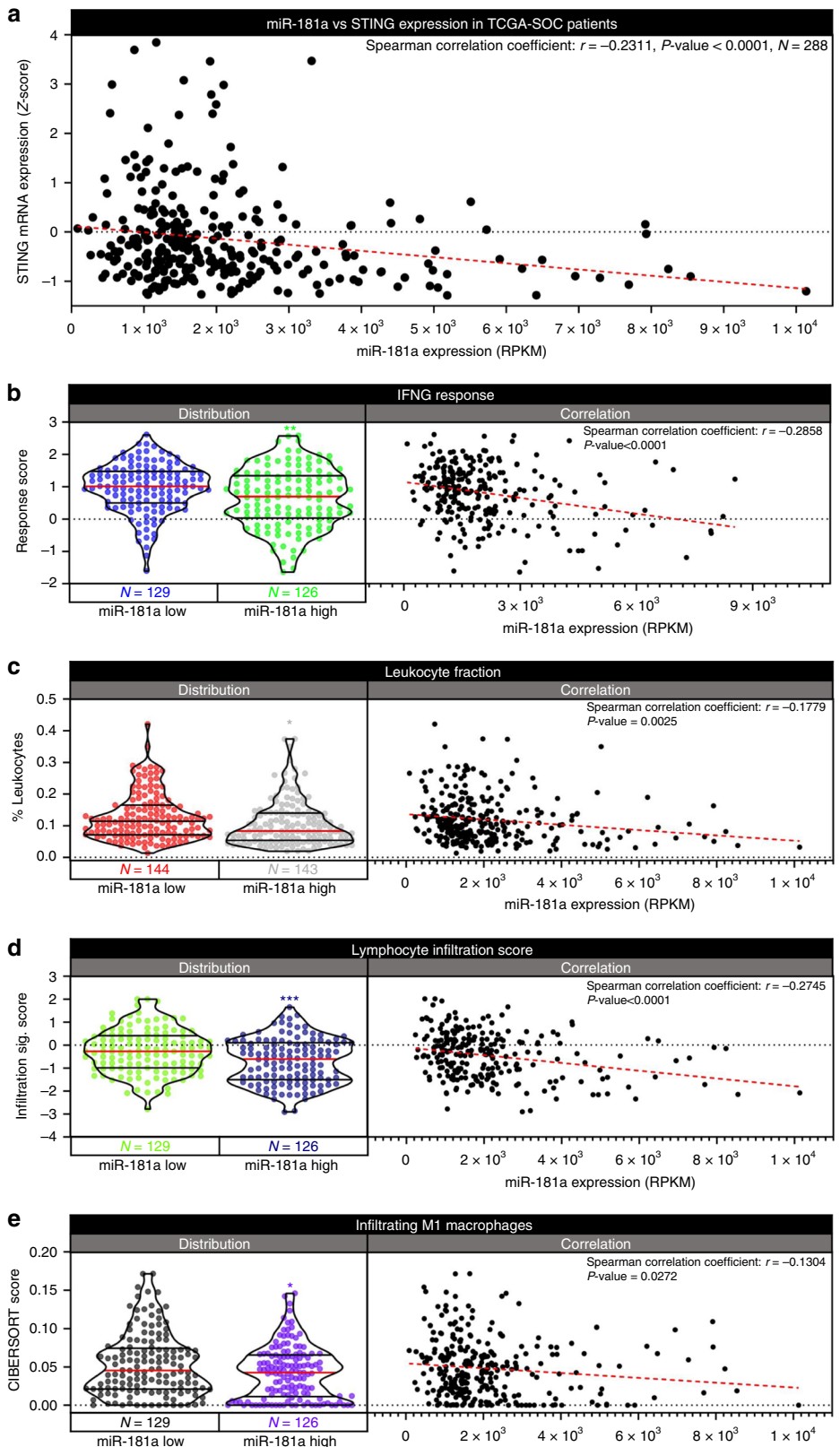

some cases used RAS activation) both of which have questionable relevance to a model reflective of HGSOC transformation in patients. Our results showed that miR-181a was able to promote tumor formation independent of viral oncoproteins. YAP signaling activation is the only known driver that has been shown to promote tumor formation as a stand-alone oncogene in non-viral oncoprotein immortalized FTSECs[18].

A unique feature of miR-181a was that it could both drive aneuploidy and generate a permissive intracellular milieu for those aneuploid cells to survive (Fig. 3 and Supplementary Fig. 4).

**Fig. 10 miR-181a inversely correlates with immune activation in HGSOC patient tumors. a** Graph of TCGA-SOC patient correlation analysis of miR-181a vs STING expression with Spearman correlation coefficient and *p* value (upper right). **b** Violin plot of IFNG Response score distribution in the miR-181a Low and miR-181a High subpopulations of TCGA-SOC patients (Left) along with correlation analysis graph of miR-181a expression vs IFNG Response score across all TCGA-SOC patients (Right). **c** Violin plot of leukocyte fraction distribution in the miR-181a Low and miR-181a High subpopulations of TCGA-SOC patients (Left) along with correlation analysis graph of miR-181a expression vs leukocyte fraction across all TCGA-SOC patients (Right). **d** Violin plot of lymphocyte infiltration score distribution in the miR-181a Low and miR-181a High subpopulations of TCGA-SOC patients (Left) along with correlation analysis graph of miR-181a expression vs lymphocyte infiltration score across all TCGA-SOC patients (Right). **e** Violin plot of Infiltrating M1 Macrophage score distribution in the miR-181a Low and miR-181a High subpopulations of TCGA-SOC patients (Left) along with correlation analysis graph of miR-181a expression vs Infiltrating M1 Macrophage score across all TCGA-SOC patients (Right). Numbers of patients in miR-181a Low and High subpopulations are located below left graph. Linear regression of correlation analysis in right graph is shown as dashed red line. Spearman's rank order correlation analysis was used for statistical analysis. Spearman correlation coefficient and *p* value for correlation analysis is shown in upper right. *$p < 0.05$, **$p < 0.005$, ***$p < 0.0005$.

Mechanistically, we show that miR-181a was able to initiate punctuated LSGI and tumorigenesis in the FTSECs by simultaneously targeting RB1 and STING. RB1 is a critical tumor suppressor in several types of cancer including HGSOC. While the role of RB1 in HGSOC progression has been investigated[24,32,35], its specific function in the early stages of FTSEC transformation and the mechanisms by which it is lost (apart from LOF mutations which account for only 15% of HGSOC patients) have yet to be elucidated. The canonical role of RB1 as a tumor suppressor is through inhibition of the G1/S cell cycle progression. However, there is increasing evidence that RB1 has a number of non-canonical tumor suppressor functions including regulating DNA damage responses, inducing senescence, activating apoptosis, and preventing aneuploidy/chromosomal instability[36–45]. RB1 has also been reported to prevent nuclear rupture events[46]. All of these phenotypes were inhibited with miR-181a overexpression, however, the profound effects on DNA damage, nuclear defects, and CNV alterations observed in the pmiR-181a cells were not maintained with targeted inhibition of RB1. The most plausible explanation for this difference would be that RB1 is not the sole miR-181a target responsible for the observed phenotypes and that other targets contribute to the propagation of these cells with a high degree of genomic instability.

Given that interferon signaling has a number of tumor suppressive effects that can be tumor cell intrinsic as well as immune regulatory[47,48], as well as the fact that it was the top significantly altered pathway in FT pmiR-181a cells, we hypothesized that miR-181a was also acting to disrupt the tumor suppressive effects of interferon signaling[47,48]. We found that STING was downregulated with miR-181a overexpression and rescued with the addition of the miR-181a antagomiR. STING is an integral component of interferon signaling that acts as a critical mediator in the sensing of cytoplasmic dsDNA, and induces interferon signaling that can lead to cell death[49]. While STING was initially investigated as an inducer of interferon signaling in response to cytoplasmic viral dsDNA, recent evidence has shown that STING can also induce interferon signaling in response to host cell genomic DNA located in the cytoplasm as a result of nuclear rupture and mitotic defects. We were able to show that miR-181a targets STING (Fig. 8) and miR-181a mediated STING inhibition could prevent cell death upon exposure to the STING agonist cGAMP. Our results also showed that the effects of miR-181a targeting STING in the FTSECs may be applicable to cell-extrinsic mechanisms of STING mediated tumor suppression such as immune cell recruitment and activation. This is plausible given that miR-181a overexpression decreased secretion of CXCL10, an important chemokine necessary for immune cell recruitment (and subsequent activation) to the tumor microenvironment (Supplementary Fig. 9C). It will be important in future studies to parse out the relative contributions of cell-intrinsic and cell-extrinsic effects of miR-181a-STING targeting in transforming FTSECs using immune competent models that fully recapitulate the microenvironment of HGSOC transformation.

Furthermore, we found that loss of STING increased phenotypes associated with transformation including cell proliferation, survival, and anchorage independent growth. Loss of STING also conferred protection against cell death in the event of increased LSGI (Fig. 9). These data provided further support that STING acts as a check against LSGI during HGSOC transformation, and that miR-181a targeting of STING allowed the FTSECs to bypass that check. In addition, overexpression of STING lacking a 3′UTR in the FTSECs showed that restoration of STING activity in the miR-181a cells was sufficient to arrest cells with mitotic/cytokinetic defects and aneuploidy to eventually cause cell death (Supplementary Fig. 10). Together, these data indicate that STING has additional previously unknown tumor suppressive roles beyond its function as a cytoplasmic DNA sensor, and open up the possibility of leveraging aberrations in the STING tumor suppressor pathways to develop novel early intervention strategies.

Lastly, we found that miR-181a expression inversely correlated with STING expression in patient tumors. Remarkably, we also found that tumors with high miR-181a expression had decreases in IFNγ response and validated metrics of immune cell infiltration/activation (Fig. 10). These data highlight that miR-181a targeting of STING is clinically relevant and presents a unique therapeutic opportunity. Importantly, our results establish miR-181a as a potential immune response biomarker. MiR-181a could therefore be used as marker to identify patients that may be more responsive to immunotherapy and immune reactivation strategies. In addition, these studies provide rationale for combining various immunotherapies with miR-181a targeting drugs. Given that we found miR-181a amplification significantly correlated with poor outcome across multiple different tumor types, it would be interesting to determine whether the miR-181a-STING axis is relevant in other cancers.

Our results have illuminated intriguing mechanisms that can contribute to early HGSOC development, and our data suggest that miR-181a has potential as a biomarker for early detection of HGSOC. Increased expression of miR-181a could allow pre-malignant p53 signature lesions to escape immune surveillance and promote resistance to anti-tumor immune response in STIC/HGSOC. Further studies will need to investigate in detail the immune suppressive capacity of miR-181a in immune competent experimental models of HGSOC development and recurrence. In addition, studies will need to investigate miR-181a as an early detection biomarker in prospective clinical samples such as serum or plasma taken prior to HGSOC diagnosis. One area that remains unexplored is what induces miR-181a expression in FTSECs? Some studies have shown that miR-181a can be induced in response to chronic

inflammation, as well as oxidative stress, and is present in follicular fluid[50–52].

Our studies also highlight the essential role FTSEC heterogeneity can play in determining transformation outcome. Our data show that RB1-functional status has a key role in determining the potency of miR-181a as an oncomiR. Our use of the FTSEC line FT194 (which was immortalized using viral oncoproteins to inactivate RB1) highlighted the importance of a functional RB1 in allowing miR-181a to induce transformation given that miR-181a overexpression had no effect in this line.

In sum, the combined targeting of STING and RB1 by miR-181a creates a unique situation whereby miR-181a creates an intracellular milieu conducive to tumorigenesis and LSGI. Interestingly, it has recently been reported that a subset of HGSOC patients with low RB1 expression concurrent with defective homologous recombination have prolonged survival[53–55]. This is presumably a result of these patients having increased response to platinum agents but also potential increased activation of STING-interferon signaling as the result of increased DNA damage and genomic instability. Our results show miR-181a targeting of STING could allow HGSOCs to bypass some of the clinical benefits that occur when low RB1 levels trigger STING activation.

## Methods

**Anchorage independent growth assay**. To assess anchorage independent growth in the FTSECs cells were plated under ultra-low attachment conditions at a density of 100,000 cells per well in six-well plates. To establish ULA conditions each well of the six-well plate was coated with 2 mL of 0.25% agarose in DMEMF12. The agarose layer was allowed to solidify and then cells were dispensed in 2 mL of DMEMF12 on top of the agarose layer. The cells were cultured for 8 weeks under ULA conditions. ULA colonies were quantified using light microscopy with a minimum of ten randomly dispersed images per well. All ULA assays were performed in biological triplicate. For ULA images presented in figures the images were converted to 24 bit grayscale followed by application of the GEM LUT to highlight spheres.

**β-galactosidase staining**. To assess the presence of senescent cells, cells were plated in six-well plates at 300 K cells per well. The cells were then treated with either lipofectamine vehicle or lipofectamine + cGAMP to activate STING (see STING Activation section for details). Following STING activation, the cells were fixed and stained for β-gal activity to quantify senescent cells using a Senescence β-gal Staining Kit (Cell Signaling Technology) according to the manufacturer's protocol. β-gal positive cells were quantified using light microscopy with a minimum of ten randomly dispersed images per well. All β-gal staining assays were performed in biological triplicate.

**Cell culture**. FTSEC lines FT237, FT240, and FT246 were a generous gift from the Drapkin laboratory. All three cell lines were isolated and immortalized using a combination of hTERT overexpression, TP53 knockdown, and mutant CDK4[R24C] overexpression as described in[56]. FTSECs were cultured in a 1:1 ratio of DMEM and Ham's F12 media (Corning) supplemented with L-glutamine, 15 mM HEPES, 10% FBS (v/v) (Denville Scientific), and 0.6% Penicillin/Streptomycin (v/v) (Corning). For all experiments described the cells used were not cultured beyond 15 passages.

**Cell cycle assay**. For cell cycle analysis, cells were plated in 10 cm plates at a density of 500,000 cells per plate in biological triplicate (~40% confluency) and cultured for 48 h. After 48 h, cells were trypsinized and fixed in 70% EtOH overnight. Fixed cells were washed with PBS and then stained with propidium iodide for 30 min at room temperature. Cell cycle for each sample was then analyzed by fluorescence activated cell sorting using an Attune NXT flow cytometer (Invitrogen) and FCS Express software.

**Clonogenic assay**. For clonogenic assays cells were plated in biological triplicate in six-well plates at a density of 1000 cells/well and were cultured in DMEMF12 for 10 days to allow colonies to form. After 10 days, cells were washed with 1X PBS and plates were allowed to air dry. After drying, the cells were fixed with 10% methanol, 10% acetic acid (v/v) in dH2O for 1 h. After fixing, the cells were stained with 5% (w/v) crystal violet in 100% methanol overnight. Following staining the plates were rinsed and imaged. Colonies in each well were quantified using ImageJ software.

**Dead cell assay**. FTSECs were treated with either lipofectamine vehicle or cGAMP as described in the STING activation section. After 24 h of treatment, the cells were treated with NucGreen Dead 488 ReadyProbes reagent according to the manufacturer's protocol. The supernatant was collected from each of the wells and the cells were trypsinized and pooled with their respective supernatant. The cells were then spun down and resuspended in 100 μL followed by quantification of GFP + dead cells using the Countess II FL automated cell counter. Assays were performed in biological triplicate.

**ELISA assay**. For the CXCL10 ELISA assay, FTSECs were treated with either lipofectamine vehicle or cGAMP as described in the STING activation section. After 24 h of treatment, media from each cell line and condition was collected and assayed by ELISA for CXCL10 using the Thermo Fisher IP-10 Human ELISA Kit (cat. # KAC2361) according to the manufacturer's instructions.

**FT237 microarray and data analysis**. Global mRNA expression levels were profiled in biological triplicate in the FT237 pscram-miR, p181a, and p181a-antimiR cells using the Clariom S Human microarray assay (Applied Biosystems) and all data are available via GSE150909. For GSEA either FT237 p181a or FT237 p181a-antimiR was compared with FT237 pscram-miR using the recommended default parameters. An FDR q-value < 0.25 was used as the cut-off for significance in all GSEA analysis. For IPA a core analysis of either FT237 p181a or FT237 p181a-antimiR vs FT237 pscram-miR was run using the recommended default parameters. IPA calculated p values (right tailed Fisher's exact test) of p < 0.05 were used as the cut-off for significance in all analyses.

**FT237 SNP array and data analysis**. Copy number variants were profiled in biological duplicate in the FT237 pscram-miR, pmiR-181a, pmiR-181a + antimiR, and pshRB1 cells using the Infinium CytoSNP-850K v1.2 BeadChip (Illumina) and all data are available via GSE. For CNV analysis, processed data were analyzed using CNV partition according to the default parameters.

**Immunocytochemistry**. *Analysis of nuclear shape*: For analysis of nuclear shape FTSECs were seeded in 4 well chamber slides at a density of 70,000 cells per well (~70% confluent) and were allowed to grow for 48 h. The cells were washed with 1X PBS and then fixed with 4% paraformaldehyde for 10 min. Cells were washed with 1X PBS three times for 5 min each followed by permeabilization with 0.01% (v/v) Triton X100 (Sigma) for 10 min. The cells were washed 3X with 1X PBS for 5 min each followed by incubation with ActinGreen 488 ReadyProbes Reagent (Invitrogen) for 30 min. The cells were then washed 3X with PBS for 5 min each followed by mounting with Prolong Diamond w/DAPI. Cells were imaged using a Leica DMI6000 inverted microscope. Circularity of nuclei was calculated using ImageJ software. Cells were imaged in biological triplicate with each biological replicate consisting of a minimum of 50 nuclei analyzed.

*PML staining*: For PML staining cells were seeded in four-well chamber slides at a density of 70,000 cells per well (~70% confluent) and were allowed to grow for 48 h. The cells were washed with 1X PBS and then fixed with 4% paraformaldehyde for 10 min. Cells were washed with 1X PBS three times for 5 min each followed by permeabilization with 0.01% (v/v) Triton X100 (Sigma) for 10 min. The cells were washed 3X with 1X PBS for 5 min each followed by blocking in 3% bovine serum albumin for 1 h. The cells were then incubated with anti-PML primary antibody (Abcam cat. # ab96051 1:500 dilution) overnight. The cells were then washed 3X with PBS for 5 min each followed by incubation with anti-rabbit Dylight488 conjugated secondary antibody (Thermo Fisher cat. #35553 1:100 dilution) in 3% BSA for 1 h. After incubation with the secondary antibody, the cells were then washed 3X with PBS for 5 min each followed by counterstaining with ActinRed 555 ReadyProbes reagent (Invitrogen) for 30 min. After counterstaining, the cells were then washed 3X with PBS for 5 min each followed by mounting with Prolong Diamond mountant w/DAPI. Cells were imaged using a Leica DMI6000 inverted microscope. LAS-X imaging software was used to analyze images for cytoplasmic PML + cells. Cells were imaged in biological triplicate with each biological replicate consisting of a minimum of 30 cells analyzed.

**Immunohistochemistry**. Immunohistochemistry was performed on consecutive 4-μm-thick sections of formalin fixed and paraffin embedded FTSEC mouse tumors. Antigen retrieval and staining for gamma H2AX (clone and manufacturer?) was performed as previously described in ref. [57]. Staining for PAX8 (Roche, clone MRZ-50) and Ki-67 (Roche, clone 30-9) was performed on an automated BenchMark ULTRA staining module (Ventana) using ULTRA CC1 cell conditioning for antigen retrieval and the OptiView DAB Detection Kit (Ventana). All slides were analyzed in a blinded fashion by a gynecological pathologist.

**In vivo tumor formation assays**. All in vivo experiments were conducted according to protocols approved by the Animal Research Committee at Case Western Reserve University.

*Subcutaneous tumor formation*: Overall, $2 \times 10^7$ cells were injected subcutaneously in a 1:1 mixture of DMEMF12 media and Matrigel matrix (Corning). For each group (FT237 p000, FT237 p181a, FT237 p181a-antimiR) a

total of five mice were used per group with two injection sites per mouse for a total of ten tumor injections per group. Tumors were measured biweekly for 25 weeks. At 25 weeks all mice were euthanized and tumors were recovered for end-point measurements and histological analysis.

*Intraperitoneal (IP) tumor formation*: Overall, $2 \times 10^7$ cells were injected intraperitoneally in a 1:1 mixture of DMEMF12 media and Matrigel matrix (Corning). For each group (FT237 p000, FT237 p181a, FT237 p181a-antimiR) a total of ten mice were used per group with one injection per mouse for a total of ten tumor injections per group. Mouse weights were monitored biweekly for 22 weeks. At 22 weeks, all mice were euthanized and IP nodules were recovered for end-point measurements and histological analysis.

**Live cell imaging**. *Live cell imaging of nuclear shape*: Overall, 100,000 cells were plated in a six-well plate and allowed to grow overnight. The cells were then treated with CellLight Nucleus-GFP BacMam 2.0 (Thermo Fisher) according to the manufacturer's instructions. After treatment, the cells were imaged using a Leica DMI6000 inverted microscope with live cell incubation chamber for 48 h with images taken every 2 min. LAS-X imaging software was used for analysis of nuclear shape before and after cell division. Normal nuclei were classified as nuclei without any noticeable major defects in shape. Abnormal nuclei were classified as nuclei with noticeable defects in nuclear shape (i.e., binucleate, micronuclei, multilobuled nuclei etc.). Cells were imaged in biological triplicate with each biological replicate consisting of a minimum of 20 cell divisions analyzed.

*Live cell imaging of mitotic/cytokinetic defects*: Overall, 100,000 cells were plated in a six-well plate and allowed to grow overnight. The cells were then treated with CellLight H2B-GFP BacMam 2.0 (Thermo Fisher) according to the manufacturer's instructions. After treatment, the cells were imaged using a Leica DMI6000 inverted microscope with live cell incubation chamber for 48 h with images taken every 2 min. LAS-X imaging software was used for analysis of MITOC abnormalities. Cells were imaged in biological triplicate with each biological replicate consisting of a minimum of 20 cell divisions analyzed.

**Lentiviral transduction**. For stable transduction of FTSECs with miR-181a over-expression, miR-181a antagomiR, and RB1 shRNA lentiviral vectors HEK 293T cells were cotransfected with the lentiviral expression vector along with pPACKH1 Packaging Plasmid Mix (System Biosciences) according to the manufacturer provided protocol. FTSECs (passage 5 or earlier) were then transduced with the lentivirus according to the manufacturer provided protocol. For selection of the FTSECs transduced with miR-181a overexpression (both p000 and p181a vectors) RFP flow cytometry was used. For cells transduced with miR-181a antagomiR (both cmiR and antimiR vectors) hygromycin selection was used. Post transduction, cells were selected with 500 μg/mL hygromycin in DMEMF12 for 48 h. After recovery, selection was maintained by treatment with 500 μg/mL hygromycin in DMEMF12 for 48 h every three passages. miR-181a antagomiR transduction efficiency was assessed by qRT-PCR. For selection of the FTSECs transduced with RB1 shRNA GFP flow cytometry was used. All cell lines selected with flow cytometry were analyzed every 3–5 passages to make sure that the RFP or GFP + population was >95% with resorting done as needed.

**Luciferase reporter assays**. 3′UTR luciferase reporter assays were carried out using the pEZX-MT06 dual-luciferase reporter construct containing either an RB1 3′UTR firefly luciferase reporter or STING 3′UTR firefly luciferase reporter. Each 3′UTR reporter also contained a synthetic renilla luciferase reporter under the control of a constitutive promoter (Genecopoeia) in the same construct. FTSECs were transfected with 1000 ng of the dual RB1 or STING 3′UTR/renilla house-keeping plasmid using lipofectamine 2000 (Thermo Fisher Scientific). Cells were harvested and lysed after 24 h of transfection using the Promega Dual-Luciferase® Reporter Assay System. Luciferase activity was then measured using a GloMax®-Multi Detection System (Promega). Firefly luciferase activity was normalized to the housekeeping renilla luciferase activity. Mutant 3′UTR reporter plasmids were generated using site directed mutagenesis targeted to the predicted miR-181a seed sequence binding site within each 3′UTR.

**MTT cell viability assay**. For cell viability assays cells were plated at a density of 50,000 cells per well in a six-well plate in biological triplicate and were allowed to grow for 10 days. Viability was assessed at the indicated time points by treating the cells with MTT for 2 h. The MTT was then extracted by aspirating the media from each well and adding 300 μL of n-propanol. MTT absorbance at $\lambda = 600$ nm was measured for each sample.

**miRNA target site prediction**. Searches for potential miR-181a targets were done using Targetscan and miRwalk algorithms. Targets were then prioritized based on target relevance to the miR-181a transformation phenotype, number of binding sites within the 3′UTR, and seed sequence complementarity.

**Nuclear permeability assay**. 100,000 cells were plated in a six-well plate and allowed to grow overnight. The cells were then treated with CellLight Nucleus-GFP BacMam 2.0 (Thermo Fisher) according to the manufacturer's instructions. After treatment, the cells were imaged using a Leica DMI6000 inverted microscope with live cell incubation chamber for 48 h with images taken every 2 min. LAS-X imaging software was used for analysis of nuclear shape before and after cell division. Each image field was quantified for nuclear ruptures (frequency of ruptures for individual cells during the 48 h viewing time and percentage of cells with at least 1 rupture event out of the total population in the image field). Changes in nuclear and cytoplasmic GFP brightness were quantified using BZ-X Analyzer software. Cells were imaged in biological triplicate with each biological replicate consisting of two image fields (technical replicates) with ~20–30 cells per field.

**Plasmids**. The following plasmids were used for stable lentiviral transduction of FTSECs: pPACKH1-GAG, pPACKH1-REV, pVSV-G (Systems Biosciences), pLV-[mir-control], pLV-[mir-181a] (Biosettia), pEZX-AM04-cmiR, pEZX-AM04-anti-miR-181a (Genecopoeia), pshRB1 TRCN0000295842, pLKO.1-puro Non-Target shRNA (Sigma). For 3′UTR luciferase assays pEZX-MT06-RB1 HmiT021640-MT06 and pEZX-MT06-TMEM173 HmiT100627-MT06 (Genecopoeia) was used.

**Quantitative reverse transcriptase polymerase chain reaction**. Both mRNA and miRNA were isolated using a total RNA isolation kit (Norgen). To determine expression levels of miR-181a, miR-181b, and miR-16 100 ng of total RNA was converted to cDNA using a Taqman Reverse Transcription Kit and Taqman miRNA specific primers (ABI). The miRNA cDNA was then PCR amplified using a Roche Lightcycler II real-time PCR machine along with miRNA specific Taqman probes and ABI universal UNG master mix. To determine mRNA expression levels 1000 ng of total RNA was reverse transcribed into cDNA using the Transcriptor Universal cDNA Synthesis Kit (Roche). cDNA was then PCR amplified using a Roche Lightcycler II real-time PCR machine along with gene specific PCR primers and LightCycler® 480 SYBR Green I Master Mix (Roche). Primers used are listed in Supplementary Table 1.

**Statistics and reproducibility**. All statistical analyses were performed using Graphpad Prism 8.4.2. Detailed statistical results for all data are given in Supplementary Tables 3–20. All experiments were performed in biological triplicate unless otherwise stated.

**STING activation**. For STING activation assays cells were plated in six-well plates at 300 K cells/well. Cells were then treated with either lipofectamine vehicle or 10 μg/well of the endogenous STING ligand 2′, 3′ cyclic GMP-AMP (cGAMP) (Cayman Biochemical) plus lipofectamine. After 24 h of treatment, the cells were either harvested or analyzed for the appropriate downstream assays.

**Survival and correlation analysis of TCGA-SOC patients**. Survival and correlation analysis of TCGA-SOC patients and patient subpopulations was performed using Graphpad Prism Software. Differences in survival curves were computed using the log-rank test. Correlation analysis of patient expression values were computed using Spearman's rank order correlation.

**Western blot**. Total protein lysates were prepared from FTSEC pellets using RIPA buffer supplemented with PhosSTOP phosphatase inhibitor and cOmplete protease inhibitor cocktail (Roche). Protein concentration of FTSEC lysates was determined using BCA assay. Western blots were run as described previously[22]. Primary and secondary antibodies used are listed in Supplementary Table 2.

**Reporting summary**. Further information on research design is available in the Nature Research Reporting Summary linked to this article.

## Data availability
The data that support the findings of this study are available from the corresponding author upon reasonable request. Microarray and SNP array data generated during this study are publicly available in the Gene Expression Omnibus Database under the accession number GSE150909.

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

## Acknowledgements

The authors would like to thank several core facilities within the Case Comprehensive Cancer Center (P30CA043703) including Michael Sramkoski from the Cytometry and Microscopy Core for assistance with flow cytometry and live cell imaging, Dr. Martina Veigl, Patrick Leahy, and the CWRU Integrated Genomics Shared Resource for assistance in performing microarray experiments and data analysis and the

Athymic Animal and Preclinical Therapeutics Core. The authors would like to thank the University of Michigan Biomedical Research DNA Sequencing Core Facility for assistance with performing SNP array and data analysis (P30CA046592). This work was supported by grants from The National Cancer Institute, R01CA197780 (A.D.), Department of Defense, OC150553 (A.D.), The Young Scientist Foundation (A.D.), and CWRU Pharmacology Department MTTP Training Grant (T32GM008803-11A1), Developmental Pilot (M.K.), CTSA/KL2, TR0002547 (S.A.), and NIH/NCI (Pass-through Roswell Park Cancer Institute, P50CA159981). In addition, we would like to thank all the generous donors and Foundations who have supported the DiFeo lab and strive to improve the outcomes of ovarian cancer patients including Norma C. and Albert I. Geller, The Silver Family Foundation and Jacqueline E Bayley (JEB) Foundation.

## Author contributions

M.K. generated and characterized all stable cell lines. M.K. and A.D designed and analyzed all experiments and data. M.K. contributed the data in Figs. 1–10 and Supplementary Figs. 1–12. L.J.K. contributed to the data in Figs. 6 and 7. R.A. contributed to the data in Fig. 9. S.Sekhar contributed to the data in Supplementary Fig. 10. J.M contributed to the data in Fig. 8. M.D. contributed to Fig. 5. S.Skala and S.A. contributed to the data in Fig. 2. M.K. and A.D. wrote the main manuscript text. R.D. provided the FTSE cell lines. S.Skala and S.A. provided pathology support and analyzed histological and immunohistochemical slides of miR-181a induced tumors. All authors reviewed the manuscript.

## Competing interests

The authors declare no competing interests.
