## [Peer Review File · Nature Communications]

Reviewers' comments:

Reviewer #2 (Remarks to the Author):

In the manuscript "miR-181a initiates and perpetuates oncogenic transformation through the regulation of innate immune signaling." Knarr et al. show that miR-181a is able to transform fallopian tube secretory epithelial cells through the inhibition of RB1 and drive a cell-protective inhibition of the stimulator of interferon genes (STING). miR-181a inhibition of RB1 leads to profound nuclear defects, genomic instability, nuclear rupture, and DNA damage. miR-181a directly downregulates STING and prevents cell death and suppresses activation of the immunosurveillance machinery. Overall the manuscript is well-written and the results are very clear. Especially, the inverse correlation between STING mRNA expression and miR-181a imply the clinical relevance of this interaction in patient samples. The implications of these findings are important for novel patient selective biomarkers as well as a therapeutic target for HGSOc. Here are major/minor concerns that should be addressed to support these findings:

1. In this manuscript, the authors showed that miR-181a inhibits RB1 and leads to nuclear defects, genomic instability, nuclear rupture, and DNA damage. However, in Supplemental Figure 7, RB1 alone could not account for the full increase in the propagation of cells with aneuploidy and nuclear defects as observed in the FT237 pmiR-181a cells. There are maybe other critical targets among the 25 potential candidates to support miR-181a oncomir activity.
2. miR-181a increases nuclear rupture and mitotic defects in FTSECs and promotes DNA damage and drives genomic instability. Recently, it has been reported that micronuclei occur after genomic instability and mis-segregation of DNA during cell division induces cGAS to localize to the micronuclei to generate cGAMP and induce STING signaling. The authors need to check the level of micronuclei in the cytoplasm after miR-181a overexpression. Additionally, the authors need to verify if cGAS is expressed in the FTSECs. If there is no cGAS, STING should be not activated in the FTSECs cells regardless of miR-181a existence. Is there active cGAS/STING signaling in the FTSECs cells?
3. In Figure 5N, the authors showed that STING activation by cGAMP induced a significant decrease in cell viability and increased cell death using MTT assay and % of dead cell using a cell-impermeant green fluorescence. Are these results STING dependent? The authors claimed that miRNA-181a decreased STING expression and prevented the cell death by inhibiting STING signaling, however there is no evidence the cell death occurs in a STING dependent manner. In addition, it's known that STING signaling is responsible for cell senescence. The authors need to evaluate whether miR-181a targets STING and inhibits cellular senescence.
4. What is the effect of STING knockdown to prevent cell death and induce cell viability or proliferation? Additionally, does STING knockdown increase tumor formation compare to miR-181a overexpression?
5. Figure 5 is too crowded.

Reviewer #3 (Remarks to the Author):

This is an interesting paper describing the role of miR-181a in oncogenic transformation. The authors identified miR-181a as an oncogenic microRNA, which transforms fallopian tube secretory epithelial cells, leading to profound nuclear defects, genomic instability, nuclear rupture, and down regulation of interferon response. The authors identified several candidate targets of miR-181a, including RB1 and STING.

These are novel and interesting findings.

However, there are several concerns.

Major points:

- 1) In Supplemental Figure 4 A, pscram-miR and pmiR-181a look very similar, while pmiR-181a+ anti-miR looks very different from other two. Is this correct result? Or just mislabeling of the

figure?

2) Is RB1 a direct target of miR-181a? The authors should show the binding sites of miR-181a in the 3'UTR of RB1 and test the effect of some mutations of the binding sites on the miR-181a binding to the 3'UTR. Are the binding sites evolutionarily conserved?

3) Is STING a direct target of miR-181a? The authors should show the binding sites of miR-181a in the 3'UTR of STING and test the effect of some mutations of the binding sites on the miR-181a binding to the 3'UTR. Are the binding sites evolutionarily conserved?

4) In Figure 5H, the reduction of RB1 protein expression in miR-181a cells is very mild and I am not convinced that this level of reduction of RB1 protein can cause the profound effects on nuclear defects, genomic instability, nuclear rupture, or cell proliferation. The authors should test whether miR-181a can affect nuclear defects, genomic instability, nuclear rupture, and cell proliferation in RB1-negative cells. If they see the effects in RB1-negative cells, then RB1 will not be a relevant target for these effects.

5) If the authors want to prove that the effects of miR-181a on impaired IFN response is due to inhibition of STING by miR-181a, the authors should test if re-expression of miR-181a-resistant STING can reverse the impaired IFN response in miR-181a cells. This is a critical point.

Reviewer #4 (Remarks to the Author):

Knarr et al. presented an interesting study on the oncogenic role of miR-181a in the early carcinogenesis of HGSOV from FTE pre-cursors via suppressing RB1 AND regulating innate immune signaling at the same time. In general, the presentation of the data is clear with high quality. The manuscript is well prepared. While this is an intriguing hypothesis with some convincing evidence, however, there are still several unsolved issues.

The authors used three FTE samples to show the consistency of the mechanisms. However, this is where the authors could explore further the heterogeneity of the role of miR-181a.

From the TCGA data presented in Figure 6, the correlation coefficient of miR-181 with STING was marginal (-0.23; despite being significant). From the dot plot, it was apparent that there were samples which showed better correlation than others, so there was definitely heterogeneity of how miR-181a would be the driving mechanism. This was reflected in the varied phenotypes shown among the three FTE samples that FT237 had the most consistent/significant results while FT240 and FT246 showed variable cell morphologies, RB/STING suppression following miR-181a expression. FT237 showed a mesenchymal phenotype while FT240/246 showed a hybrid phenotype. Could this mean that there would be different functional consequences of miR-181 expression in FTEs from different donors? Would these varied consequences contribute differently in the oncogenic property among different patients? If yes, then what would be the "host factor" which could explain the variable degree of RB/STING down-regulation? Was there a dosing effect of miR-181a and the RB/STING down-regulation which could explain the varied phenotypes? It is known that cells with different morphologies would have different miRNA regulatory networks. Whether these would further impact on the miR-181a function (ceRNAs for RB or STING?) needs to be clarified. I would appreciate the authors to explore further from the existing data.

REVIEWERS' COMMENTS:

Reviewer #2

Q1: In the manuscript "miR-181a initiates and perpetuates oncogenic transformation through the regulation of innate immune signaling." Knarr et al. show that miR-181a is able to transform fallopian tube secretory epithelial cells through the inhibition of RB1 and drive a cell-protective inhibition of the stimulator of interferon genes (STING). miR-181a inhibition of RB1 leads to profound nuclear defects, genomic instability, nuclear rupture, and DNA damage. miR-181a directly downregulates STING and prevents cell death and suppresses activation of the immunosurveillance machinery. Overall the manuscript is well-written and the results are very clear. Especially, the inverse correlation between STING mRNA expression and miR-181a imply the clinical relevance of this interaction in patient samples. The implications of these findings are important for novel patient selective biomarkers as well as a therapeutic target for HGSOC.

RESPONSE: We appreciate the support of the manuscript by the reviewer and have now addressed all the major/minor concerns raised as detailed below.

Q2: In this manuscript, the authors showed that miR-181a inhibits RB1 and leads to nuclear defects, genomic instability, nuclear rupture, and DNA damage. However, in Supplemental Figure 7, RB1 alone could not account for the full increase in the propagation of cells with aneuploidy and nuclear defects as observed in the FT237 pmiR-181a cells. There are maybe other critical targets among the 25 potential candidates to support miR-181a oncomir activity.

RESPONSE: The reviewer's point is well taken and we have now performed additional studies showing that miR-181a targeting of STING also contributes to phenotypes that increase genomic

instability such as nuclear rupture and increased DNA damage (Figure 7). With respect to other targets, it is well known that miRNAs can exert their phenotypic effects by regulating multiple targets however through shRNA phenocopy studies and STING re-expression studies we confirmed that miR-181a targeting of these 2 proteins was driving the observed phenotypes. The other 23 targets identified could still be driving the miR-181a induced effects however the phenotypic effects of target loss were either understudied or less clearly linked to the miR-181a driven phenotype we observed in the FTSECs. It is possible that miR-181a additionally drives transformation and increased genomic instability by targeting some of the additional 23 targets presented and this could be investigated in future studies. However, our data suggests that the majority of the phenotype driven by miR-181a is mediated through miR-181a targeting of RB1 and STING.

Q3. miR-181a increases nuclear rupture and mitotic defects in FTSECs and promotes DNA damage and drives genomic instability. Recently, it has been reported that micronuclei occur after genomic instability and mis-segregation of DNA during cell division induces cGAS to localize to the micronuclei to generate cGAMP and induce STING signaling. The authors need to check the level of micronuclei in the cytoplasm after miR-181a overexpression. Additionally, the authors need to verify if cGAS is expressed in the FTSECs. If there is no cGAS, STING should be not activated in the FTSECs cells regardless of miR-181a existence. Is there active cGAS/STING signaling in the FTSECs cells?

RESPONSE: The reviewer is correct to state that cGAS needs to be expressed in the FTSECs in order for genomic instability to activate the cGAS-STING interferon response. We have included additional experimental data where we show that the frequency of micronuclei is increased in FTSECs that overexpress miR-181a (Supplemental Figure 2A). We also show by western blot that cGAS is expressed in our FTSEC cell lines indicating that it is present to respond to genomic instability (Supplementary Figure 9D).

Q4. In Figure 5N, the authors showed that STING activation by cGAMP induced a significant decrease in cell viability and increased cell death using MTT assay and % of dead cell using a cell-impermeant green fluorescence. Are these results STING dependent? The authors claimed that miRNA-181a decreased STING expression and prevented the cell death by inhibiting STING signaling, however there is no evidence the cell death occurs in a STING dependent manner. In addition, it's known that STING signaling is responsible for cell senescence. The authors need to evaluate whether miR-181a targets STING and inhibits cellular senescence.

RESPONSE: We appreciate the reviewer's astute comments and we have now included data showing that cGAMP induced cell death is dependent on STING using both STING knockdown and overexpression cell lines. Our results presented in Figure 7F and Supplementary Figure 10B show that the cell death induced by cGAMP is dependent on STING given that cells that have targeted inhibition of STING do not respond to cGAMP treatment and conversely cells that have overexpression of STING are more sensitive to cGAMP. We have performed β -galactosidase staining in our FTSEC cells with or without STING activation and found that miR-181a does decrease cell senescence as expected but most likely by targeting RB1 rather than STING

directly. Our data show that cell senescence is not the mechanism that leads to cell death when STING is activated and thus the ability of miR-181a to prevent STING-mediated cell death in response to increased genomic instability is not dependent on blocking STING activated senescence (see Results section *miR-181a allows FTSECs to bypass genomic instability-triggered interferon response and cell death by targeting the cytoplasmic DNA sensor STING* for details).

Q5. What is the effect of STING knockdown to prevent cell death and induce cell viability or proliferation? Additionally, does STING knockdown increase tumor formation compare to miR-181a overexpression?

RESPONSE: We are very happy that the reviewer asked us to perform these assays because it has resulted in both confirmation of our hypothesis but also the discovery that STING has additional tumor suppressive roles beyond its function as a cytoplasmic DNA sensor. We have now included data from FT237 cells with stable knockdown of STING using to different shRNA hairpins. Upon confirmation of STING knockdown to levels observed in the FT237 pmiR-181a cells we determined the effects of STING knockdown alone on FTSEC transformation in comparison to miR-181a overexpression. We found that STING knockdown successfully prevented cGAMP induced cell death at a level comparable with miR-181a overexpression (Figure 7F). STING knockdown alone was sufficient to promote increased FTSEC transformation including increased cell proliferation, increased clonogenicity, and increased anchorage independent growth (Figure 7). In addition, STING knockdown was sufficient to allow the survival of cells with increased genomic instability (Figure 7). These data show that like RB1, decreasing STING expression transforms FTSECs through multiple tumorigenic phenotypes however the biological impacts varied. For example, both RB1 and STING inhibition resulted increased cell proliferation, increased clonogenicity, and increased anchorage independent growth but only the inhibition of STING allowed the survival of cells with increased genomic instability.

Q6. Figure 5 is too crowded.

RESPONSE: We have revised Figure 5 to be less crowded.

Reviewer #3

Q1: This is an interesting paper describing the role of miR-181a in oncogenic transformation. The authors identified miR-181a as an oncogenic microRNA, which transforms fallopian tube secretory epithelial cells, leading to profound nuclear defects, genomic instability, nuclear rupture, and down regulation of interferon response. The authors identified several candidate targets of miR-181a, including RB1 and STING. These are novel and interesting findings.

RESPONSE: We appreciate the support of the manuscript by the reviewer and have now addressed all the major/minor concerns raised as detailed below.

Q2: In Supplemental Figure 4A, pscram-miR and pmiR-181a look very similar, while pmiR-181a+ anti-miR looks very different from other two. Is this correct result? Or just mislabeling of the figure?

RESPONSE: We are sorry for the confusion and we have tried to better described the data in the manuscript. Briefly, the heatmaps in Supplemental Figure 4 (now Supplemental Figure 5) are organized according to the GSEA ranking algorithm using pscram-miR as the reference group. The genes are then organized in descending order according to ranked enrichment score. As indicated in Figure 4A the pmiR-181a +antimiR cells are not significantly enriched (over-represented) for the genes in the Reactive O₂ Species gene signature. Therefore, the actual pmiR-181a + antimiR gene expression values will not necessarily have any correlation (positive or negative) with the gene expression values for the pscram-miR cells.

Q3: Is RB1 a direct target of miR-181a? The authors should show the binding sites of miR-181a in the 3'UTR of RB1 and test the effect of some mutations of the binding sites on the miR-181a binding to the 3'UTR. Are the binding sites evolutionarily conserved?

RESPONSE: We have included additional data showing that RB1 is a direct target of miR-181a using a mutated RB1 3'UTR luciferase reporter and showing that miR-181a does not bind to the mutated binding site (Figure 6C). We have included diagrams of both the primary and mutated RB1 3'UTR reporter constructs (Figure 6C). In addition, we have highlighted that the miR-181a binding site in RB1 is evolutionarily conserved among mammals (Targetscan).

Q4: Is STING a direct target of miR-181a? The authors should show the binding sites of miR-181a in the 3'UTR of STING and test the effect of some mutations of the binding sites on the miR-181a binding to the 3'UTR. Are the binding sites evolutionarily conserved?

RESPONSE: We have included additional data showing that STING is a direct target of miR-181a. We mutated the miR-181a binding site in the STING 3'UTR of our luciferase reporter and show that miR-181a does not bind to the mutated binding site (Figure 7B). We have included diagrams of both the primary and mutated RB1 3'UTR reporter constructs (Figure 7B). In addition, we have highlighted that the miR-181a binding site in STING is evolutionarily conserved among mammals (Targetscan).

Q5: In Figure 5H, the reduction of RB1 protein expression in miR-181a cells is very mild and I am not convinced that this level of reduction of RB1 protein can cause the profound effects on nuclear defects, genomic instability, nuclear rupture, or cell proliferation. The authors should test whether miR-181a can affect nuclear defects, genomic instability, nuclear rupture, and cell proliferation in RB1-negative cells. If they see the effects in RB1-negative cells, then RB1 will not be a relevant target for these effects.

RESPONSE: We appreciate this concern from the reviewer therefore we have performed additional experiments that show that miR-181a overexpression does not have any significant effects on nuclear defects, genomic instability, nuclear rupture, or cell proliferation in FTSECs where RB1 is functionally inactivated by SV40 large T antigen (Supplementary Figure 8). See Results section *miR-181a targets the tumor suppressor RB1 to promote transformation and knockdown of RB1 in FTSECs phenocopies miR-181a mediated transformation* for details.

Q6: If the authors want to prove that the effects of miR-181a on impaired IFN response is due to inhibition of STING by miR-181a, the authors should test if re-expression of miR-181a-resistant STING can reverse the impaired IFN response in miR-181a cells. This is a critical point.

RESPONSE: We truly appreciate this critical point that the reviewer raises thus we have performed extensive experiments to test if re-expression of miR-181a-resistant STING can reverse the impaired IFN response in miR-181a cells. Our data show that overexpression of STING lacking a 3'UTR can reverse the impaired IFN response in the miR-181a overexpressing cells (Supplementary Figure 10). STING overexpression re-sensitizes FT237 pmiR-181a to STING induced cell death in response to increased genomic instability (Supplementary Figure 10B). We show that STING overexpression inhibits proliferation and anchorage independent growth in the FT237 pmiR-181a cells (Supplementary Figure 10 C-D). We also show that STING overexpression causes the FT237 pmiR-181a cells to undergo increased cell death in response to genomic instability and aneuploidy (Supplementary Figure 10E-G).

Reviewer #3

Q1: Knarr et al. presented an interesting study on the oncogenic role of miR-181a in the early carcinogenesis of HGSOc from FTE pre-cursors via suppressing RB1 AND regulating innate immune signaling at the same time. In general, the presentation of the data is clear with high quality. The manuscript is well prepared. While this is an intriguing hypothesis with some convincing evidence, however, there are still several unsolved issues.

RESPONSE: We appreciate the support of the manuscript by the reviewer and have now tried to address all the major/minor concerns raised as detailed below.

Q2: The authors used three FTE samples to show the consistency of the mechanisms. However, this is where the authors could explore further the heterogeneity of the role of miR-181a.

RESPONSE: We appreciate that heterogeneity is a major concern in all cancer research and we have tried to address that through the use of numerous FTE samples. We have now included an additional FT line FT194 which is unique to the 3 other lines given that it has been immortalized using viral oncoproteins and RB1 inactivated. Through the use of these lines it is clear that the oncogenic role of miR-181a is mediated through RB1 given that overexpression of miR-181a in the RB1 inactivated FT line had no effect. This would suggest that in patients that already have loss of RB1 would not be affected by miR-181a induction.

Q3: From the TCGA data presented in Figure 6, the correlation co-efficient of miR-181 with STING was marginal (-0.23; despite being significant). From the dot plot, it was apparent that there were samples which showed better correlation than others, so there was definitely heterogeneity of how miR-181a would be the driving mechanism. This was reflected in the varied phenotypes shown among the three FTE samples that FT237 had the most consistent/significant results while FT240 and FT246 showed variable cell morphologies, RB/STING suppression following miR-181a expression. FT237 showed a mesenchymal phenotype while FT240/246 showed a hybrid phenotype. Could this mean that there would be different functional consequences of miR-181 expression in FTEs from different donors?

RESPONSE: Yes, the functional consequences of any miRNA including the ability of miR-181a to transform FTSECs are not universal. The ability of a miRNA to promote a phenotype is dependent on cell context including a number of factors such as transcriptional control of the miRNA, post-transcriptional regulation of the miRNA, target abundance, the actual propensity for a miRNA to bind a target, and the downstream effects caused by the miRNA binding to a particular target. This means that in the FTE from different patients increased miR-181a expression may have a tumor suppressor effect, no effect, or an oncogenic effect depending on the factors mentioned above. Our data now shows that given that the FT194 line which has inactivation of RB1 showed that increased miR-181a expression had no effect. Nonetheless our data does show that miR-181a has a recurrent and consistent role as a pro-tumorigenic miRNA as evidenced by the fact that it was able to transform FTSECs collected from 3 different donors.

Q4: Would these varied consequences contribute differently in the oncogenic property among different patients? If yes, then what would be the "host factor" which could explain the variable degree of RB/STING down-regulation?

RESPONSE: Yes, for example our data show that miR-181a does not increase transformation in FTSECs immortalized with SV40 large T antigen. Therefore, as an example, target functional status can affect the oncogenic properties that miR-181a can exert. In patients that are RB1 null but still have wild-type STING miR-181a might confer protection to cells with aneuploidy and mitotic cytokinetic defects but might not increase other phenotypes such as anchorage independent growth. What sets miR-181a apart, and what we demonstrate in our paper, is that it has the ability to act on multiple, independent oncogenic pathways simultaneously. Furthermore, several host factors can influence the variable degree of RB1/STING down-regulation. The level of miR-181a expression is one example, patients with amplified miR-181a or factors that increase miR-181a expression (i.e. inflammation) could have a greater degree of RB1 and/or STING downregulation. As mentioned above target mutation status also affects the ability of miR-181a to decrease target levels. Patients whose FTSECs have heterozygous loss of RB1 and/or STING may be more susceptible to increased miR-181a expression due to the fact that miR-181a can act as the second "hit" to inactivate the target.

Q5: Was there a dosing effect of miR-181a and the RB/STING down-regulation which could explain the varied phenotypes?

RESPONSE: In our studies there was not enough difference in miR-181a overexpression or target knockdown between the FT240, FT237, and FT246 to make a conclusion whether there was a dosing effect between miR-181a and target knockdown. Future studies with tunable control of miR-181a overexpression levels could address this in more detail. What is clear from our results is that a threshold of a 10 to 15-fold increase in miR-181a expression is sufficient to cause RB1/STING knockdown and exert tumorigenic effects in FTSECs. This level of miR-181a upregulation or higher is what is observed in the majority of ovarian cancer patient tumors vs matched normal tissue.

Q6: It is known that cells with different morphologies would have different miRNA regulatory networks. Whether these would further impact on the miR-181a function (ceRNAs for RB or STING?). It is possible that the epithelial/mesenchymal state of the FTSECs could further impact miR-181a function.

RESPONSE: Our previous work has shown that miR-181a can promote EMT in tumor cells by targeting SMAD7 and activating TGF β signaling. In addition, TGF β and Wnt ligands are known to activate miR-181a expression therefore FTSECs cells with high TGF β and Wnt activity which tend to be more mesenchymal may be more susceptible to miR-181a induced transformation. With respect to the effects of ceRNAs, it is possible that ceRNAs could affect miR-181a driven phenotypes. For the vast majority of miRNAs, the absolute abundance of potential targets is orders of magnitude greater than the abundance of the miRNA. Thus, the effects of single or small groups of ceRNAs on miR-181a targeting RB1/STING is likely to be minimal except in cases where there is a very limited number of targets expressed for miR-181a to bind to. It is more likely that other factors that can be regulated by an epithelial/mesenchymal phenotype such as miR-181a abundance or signaling upstream/downstream of miR-181a would be the dominating contributor(s) to miR-181a function in FTSECs. Nevertheless, several ceRNAs have been shown to regulate miR-181a expression in cancer and it would be interesting to assess whether the heterogeneity observed in the patient's samples is due to ceRNA expression and future studies are warranted.

REVIEWERS' COMMENTS:

Reviewer #2 (Remarks to the Author):

The authors have well addressed most of the comments and the manuscript has been much improved. Especially, the authors have added new data regarding the direct role of STING in FTSEC transformation in response to the important major point.

Reviewer #3 (Remarks to the Author):

The authors have addressed all the concerns I raised, and the manuscript has improved.

Reviewer #4 (Remarks to the Author):

The reviewer thanks the authors for their responses to my comments. However, I don't see any of these responses being integrated into the revised manuscript. I would expect some of these concerns being addressed at least in the discussion section.

For example, in the response to Reviewer 3's Q3, the authors acknowledged that "in the FTE from different patients increased miR-181a expression may have a tumor suppressor effect, no effect, or an oncogenic effect depending on the factors mentioned above" and further provided evidence on the RB-inactivated FT194 that miR-181a had no effect in tumorigenesis. Clearly, the heterogeneity of FTE is an issue here. In their response to the same Reviewer's Q2, they did acknowledge that though miR-181a acted via RB inhibition but it had no effect in a FTE with RB inactivation. Again, I don't see the authors' efforts in highlighting these new findings in the revised manuscript. The manuscript still reads like that miR-181a has an overarching role in FTE transformation.

REVIEWERS' COMMENTS:

Reviewer #2

Q1: The authors have well addressed most of the comments and the manuscript has been much improved. Especially, the authors have added new data regarding the direct role of STING in FTSEC transformation in response to the important major point.

RESPONSE: We are happy that the reviewer is pleased with the additional data.

Reviewer #3

Q1: The authors have addressed all the concerns I raised, and the manuscript has improved.

RESPONSE: We are happy that the reviewer is pleased with the additional data.

Reviewer #4:

Q1: The reviewer thanks the authors for their responses to my comments. However, I don't see any of these responses being integrated into the revised manuscript. I would expect some of these concerns being addressed at least in the discussion section. For example, in the response to Reviewer 3's Q3, the authors acknowledged that "in the FTE from different patients increased miR-181a expression may have a tumor suppressor effect, no effect, or an oncogenic effect depending on the factors mentioned above" and further provided evidence on the RB-inactivated FT194 that miR-181a had no effect in tumorigenesis. Clearly, the heterogeneity of FTE is an issue here. In their response to the same Reviewer's Q2, they did acknowledge that though miR-181a acted via RB inhibition but it had no effect in a FTE with RB inactivation. Again, I don't see the authors' efforts in highlighting these new findings in the revised manuscript. The manuscript still reads like that miR-181a has an overarching role in FTE transformation.

RESPONSE: We are happy that the reviewer is pleased with the responses to their comments and we have now incorporated them into the discussion.